# Planktonic foraminifera regulate calcification according to ocean density

Stergios D. Zarkogiannis [1] ✉, James W. B. Rae [2], Benjamin R. Shipley[1] & P. Graham Mortyn[3]

Planktonic foraminifera are key contributors to the oceanic carbon cycle. In pelagic environments, carbonate production by planktonic biomineralizers regulates ocean-atmosphere carbon dioxide exchange and exports surface carbon to the deep ocean. Here we compare shell traits of three planktonic foraminifera species from the central Atlantic with a suite of environmental parameters to discern the factors underlying their variations. Our analysis revealed that calcification in foraminifera is associated with seawater density and depends on species habitat depth, whereas foraminifera bulk shell densities may serve as a seawater density proxy, regardless of species. We observe that their shell weights increased with habitat depth, enabling the living cells to adjust their overall density to match that of the surrounding liquid. This suggests that calcification in nonmotile organisms has a buoyancy regulatory function and will respond to the anthropogenically driven reductions in ocean density (oceanic rarefication), with potential consequences for the carbon cycle.

Planktonic foraminifera are marine protists that house their unicellular bodies within sophisticated calcite shells and are distributed globally in oceanic surface waters. They are relatively simple unicellular organisms that cannot actively swim; they can only control their buoyancy[1,2]. They occupy surface to subthermocline depths in the pelagic ocean, with regional differences in food and seawater properties constraining their latitudinal, temporal and depth distributions[3]. Most modern species are primarily epipelagic, although some descend into mesopelagic waters for reproduction or seasonal survival[4,5]. The exact mechanism by which planktonic foraminifera control their position in the water column to attain neutral buoyancy is not fully understood. However, observations suggest that species-specific buoyancy adjustments occur, concentrating populations of each species at specific depths[6]. Although positive buoyancy can be achieved by low-density metabolites like gasses and lipids[2,7] or osmolytes[8], negative buoyancy adjustment mechanisms remain poorly studied. The ability of foraminifera to control their sinking rates and maintain depth-specific habitats may be linked to their biomineralization processes.

Planktonic foraminifera have been thought to regulate their shell masses in response to ambient carbonate chemistry[9]. However, recent studies challenge this assumption, showing that calcification is not directly controlled by ocean acidity[10–12], suggesting that other environmental factors play a more dominant role. Since foraminifera can calcify by maintaining chemical gradients with their ambient environment[13,14], understanding what drives changes in shell mass requires considering the fundamental role of the shell itself. In addition to providing cellular support and protection from the biological, physical and chemical stresses of the ocean[15], increasing organismal density is

one of the adaptive roles of skeletal calcification, serving as part of a buoyancy regulation mechanism[16–18]. It has been proposed that foraminifera modify their shape and size, influencing their weight and overall density (i.e., the ratio between calcite and protoplasm), to adapt their hydrodynamic behaviour in response to changes in water conditions[19–21], a strategy that has also been confirmed in other plankton taxa[22]. Indeed, foraminifera were found to increase their shell mass with depth[23], demonstrating an active, biologically mediated carbon sequestration mechanism. Considering that planktonic foraminifera constitute a major fraction (~32–80%) of deep-sea carbonate sediments[24], understanding their biomineralization strategies is critical for understanding the history of the marine carbon cycle and hence the mechanisms regulating atmospheric carbon dioxide ($CO_2$) concentrations.

In this study, we conducted physiochemical analyses on the shells (300–350 μm in size) of three planktonic foraminifera species, namely *Globigerinoides ruber albus* sensu stricto (s.s.), *Trilobatus trilobus*, and *Globorotalia truncatulinoides*, collected from the central Atlantic surface sediment along the mid-Atlantic ridge (Fig. 1), spanning habitats from shallow to subthermocline depths. To investigate the factors influencing shell mass and calcification, we conducted Mg/Ca analyses to estimate species-specific apparent calcification depths and paired those with stable oxygen isotope ($\delta^{18}O$) measurements to reconstruct the surrounding seawater density. In addition to shell weights, we used X-ray microcomputed tomographic (μCT) data to examine physical traits such as skeletal percentage, thickness, volume, and bulk density (volume-normalized shell weights). Bulk shell density (BSD), as used here, is defined as the shell weight divided by the *"cell"* total volume derived from μCT imaging, which is the volume a living organism would occupy if the internal voids of the

[1]Department of Earth Sciences, University of Oxford, Oxford, UK. [2]School of Earth and Environmental Sciences, University of St Andrews, St Andrews, UK. [3]Department of Geography, ICTA, Universitat Autònoma de Barcelona, Barcelona, Spain. ✉e-mail: stergios.zarkogiannis@earth.ox.ac.uk

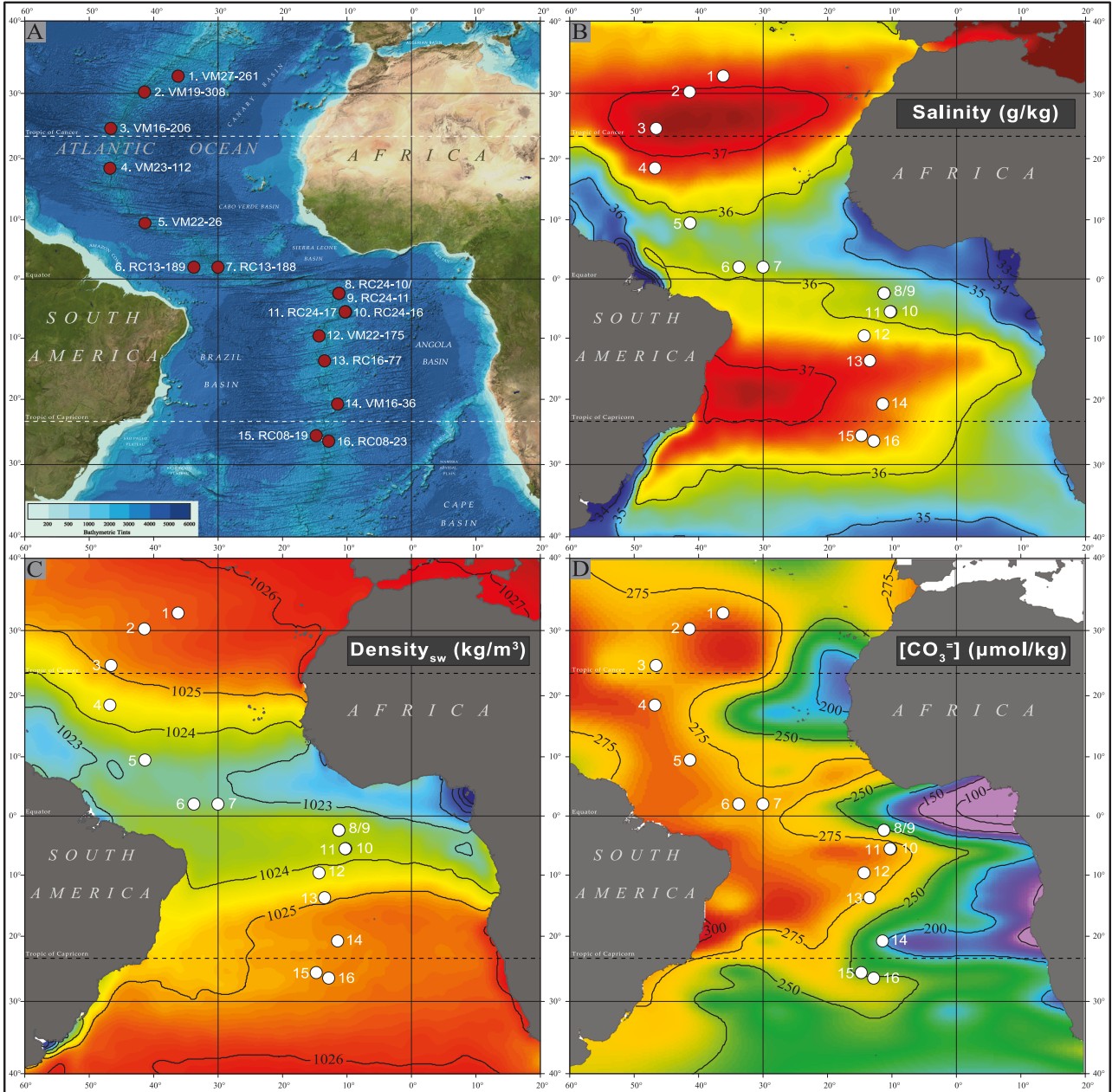

**Fig. 1 | Location of the 16 core-top samples for this study. A** Bathymetric map; **B** mean annual sea surface salinity (PSU) and **C** mean annual sea surface density data from the World Ocean Atlas 2013; **D** preindustrial surface seawater $[CO_3^{2-}]$ (µmol/kg) estimates from GLODAP v.1 data corrected for anthropogenic dissolved inorganic carbon.

(fossil) shell were fully filled with protoplasm. These six shell traits were statistically analysed via linear mixed-effects models (LMMs) and compared with geochemically reconstructed and in situ physical oceanographic parameters, as well as carbonate system properties (alkalinity, $CO_3^{2-}$), to evaluate correlations between environmental factors and calcification. Our findings indicate that shell weights and bulk shell densities (BSDs) are directly related to ambient seawater density, suggesting that planktonic foraminifera shells serve as proxies for seawater density throughout the water column. Additionally, these results imply that the ongoing ocean warming, via thermal expansion, and freshening from ice melt, which lead to oceanic rarefication (lower seawater density), may lessen the future calcification requirements of planktonic calcifiers.

## Results
### Interspecies shell response to environmental parameters
The three species exhibited significant (Kruskal-Wallis $\chi^2 > 28.7$; Benjamini and Yekuetli adjusted $p < 0.01$ for all comparisons) morphological differences across all six traits (Fig. 2). Many of these traits covary strongly (Supplementary Table 1), and across species, display similar relationships with environmental conditions.

For all six traits, the linear mixed-effects model (LMM) incorporating only seawater density consistently yielded the lowest Akaike Information Criterion (AICc) values or was indistinguishable from the best-performing model (Table 1). For all traits except test thickness, the density model performed roughly as well as the model incorporating both density and salinity (density + salinity model, Table 1). For "potential cell" volume, the density model, temperature model, salinity model, and salinity + $[CO_3^{2-}]$ models performed equally well, whereas for shell weight, the three models that incorporated density were equivalent. For test volume, the null model, incorporating only $[CO_3^{2-}]$ and temperature, performed equally well as the full model and density model. Using the density + salinity model, shell weight, BSD, test volume, test thickness, and test percentage were all positively associated with density (all $t > 2.31$, the mean effect size ranged from

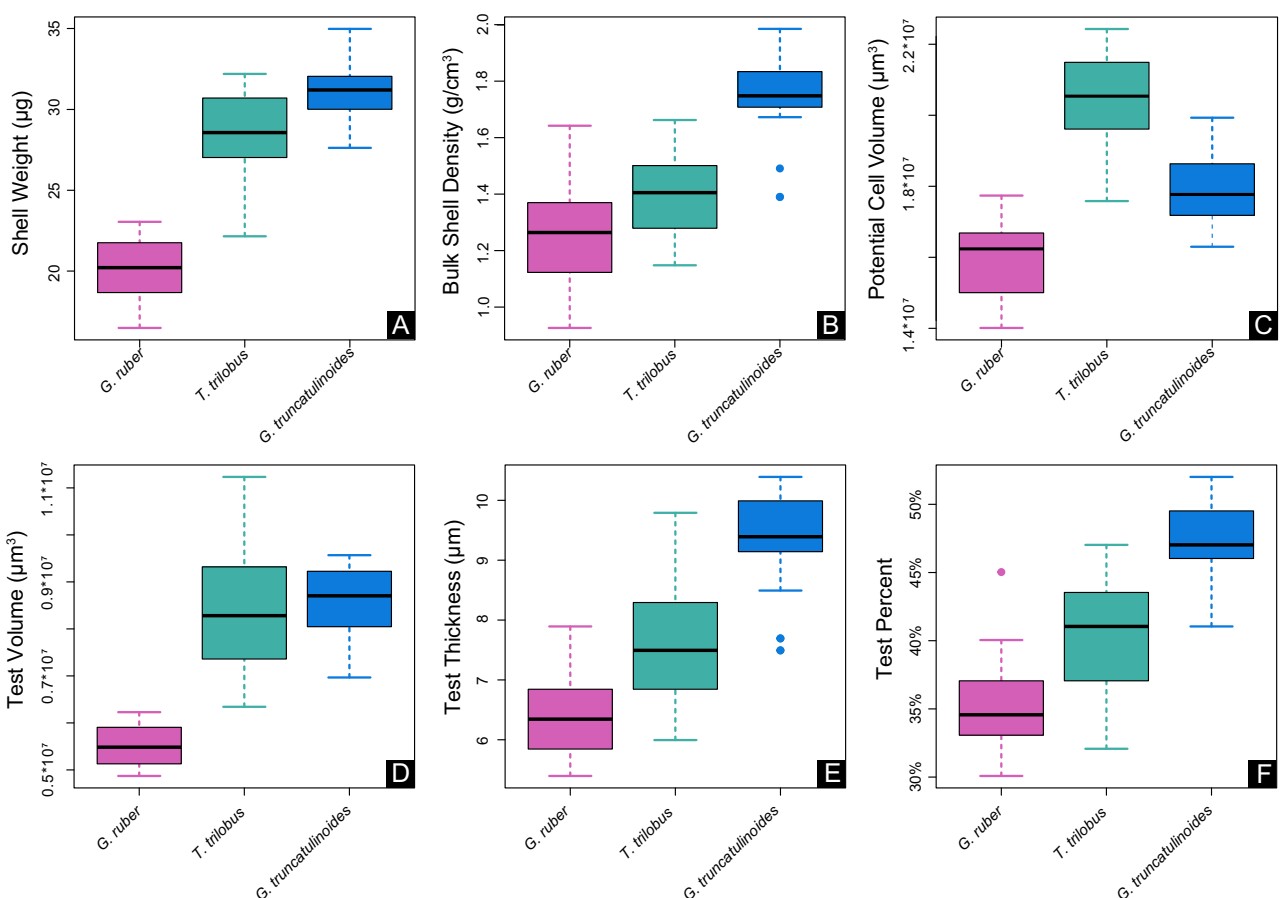

**Fig. 2 | Boxplots illustrating the morphological differences among the three species.** Plot **A** represents raw shell mass measurements, while **B** shows mass measurements normalized to **C** total volumes derived from CT data. Plots **D**–**F** depict digital traits exclusively based on CT analysis: **D** test volume, **E** test thickness, and **F** test percentage.

**Table 1 | Akaike information criterion (AICc) values for each linear mixed-effects model (LMM), with species as a random intercept**

| Model | Potential Cell Volume | Shell Weight | Bulk Shell Density | Test Volume | Test Thickness | Test Percent |
|---|---|---|---|---|---|---|
| Density | **96.20 ($R^2$ = 0.75)** | **39.30 ($R^2$ = 0.89)** | **66.36 ($R^2$ = 0.80)** | **90.86 ($R^2$ = 0.67)** | **82.04 ($R^2$ = 0.72)** | **73.34 ($R^2$ = 0.78)** |
| Salinity | **96.26 ($R^2$ = 0.71)** | 57.27 | 94.11 | 94.29 | 91.29 | 91.36 |
| Temperature | **96.77 ($R^2$ = 0.74)** | 76.28 | 111.81 | 96.38 | 102.09 | 106.90 |
| Density + Salinity | **97.79 ($R^2$ = 0.72)** | **38.13 ($R^2$ = 0.89)** | **65.27 ($R^2$ = 0.82)** | **92.59 ($R^2$ = 0.67)** | 84.35 | **73.71 ($R^2$ = 0.78)** |
| Density + [CO$_3^{2-}$] | 98.56 | **39.08 ($R^2$ = 0.89)** | **66.53 ($R^2$ = 0.81)** | 93.15 | 84.55 | **75.19 ($R^2$ = 0.78)** |
| Salinity + [CO$_3^{2-}$] | **97.89 ($R^2$ = 0.70)** | 57.85 | 93.05 | 95.47 | 90.81 | 88.20 |
| [CO$_3^{2-}$] | 98.76 | 72.22 | 109.55 | 98.80 | 101.42 | 106.40 |
| Temp + [CO$_3^{2-}$] | 98.85 | 69.38 | 110.99 | **90.67 ($R^2$ = 0.71)** | 100.60 | 104.88 |
| Full | 98.28 | **39.53 ($R^2$ = 0.90)** | 70.53 | **89.63 ($R^2$ = 0.875)** | 87.02 | 76.22 |

The best-performing model (lowest AICc) is bolded, along with all other models with ΔAICc < 2. Full =Density+Salinity+Temperature + [CO$_3^{2-}$]. All variables were centered and scaled before running the models and calculating the AICc values. Also shown are conditional pseudo-$R^2$ values, which measure the fit of the entire mixed-effects model, including both fixed and random effects; these values exceeded 0.67 for all bolded models, indicating strong overall explanatory power. AICc values for the linear models using each species individually can be found in Supplementary Table 2a–c.

$\beta = 0.40$ to $\beta = 0.96$; Supplementary Fig. 1), indicating that individuals with heavier and thicker tests were found in denser waters (Fig. 3). "Cell" volume, however, was not significantly associated with density ($\beta = -0.19$, $t = -1.00$; Supplementary Fig. 1). Although salinity showed some effect on shell weight ($\beta = 0.14$, $t = 2.00$) and BSD ($\beta = 0.13$, $t = 1.94$), it was not significantly related to other traits (all other $|t| < 1.32$). These findings challenge the common assumption that carbonate ion availability is the dominant control on calcification[9] and instead support seawater density as a primary driver.

The LMMs revealed a general effect of seawater density on the calcification of foraminifera across all three species. However, the magnitude of this effect varied between each species and trait individually (Fig. 3). The random effect term (differences in mean trait values among the species) accounted for the majority of variance in the density + salinity LMMs for shell weight (82%), "cell" volume (71%), and test volume (60%). For test thickness and test percent, the differences between traits among species were outstripped by the effect of the environmental characteristics, accounting for 13% and 9.2% of the model variance, respectively. Importantly, BSD was not significantly influenced by species differences (differences between species traits not accounted for by our environmental predictors were negligible), indicating that it serves as a species-independent proxy for seawater density.

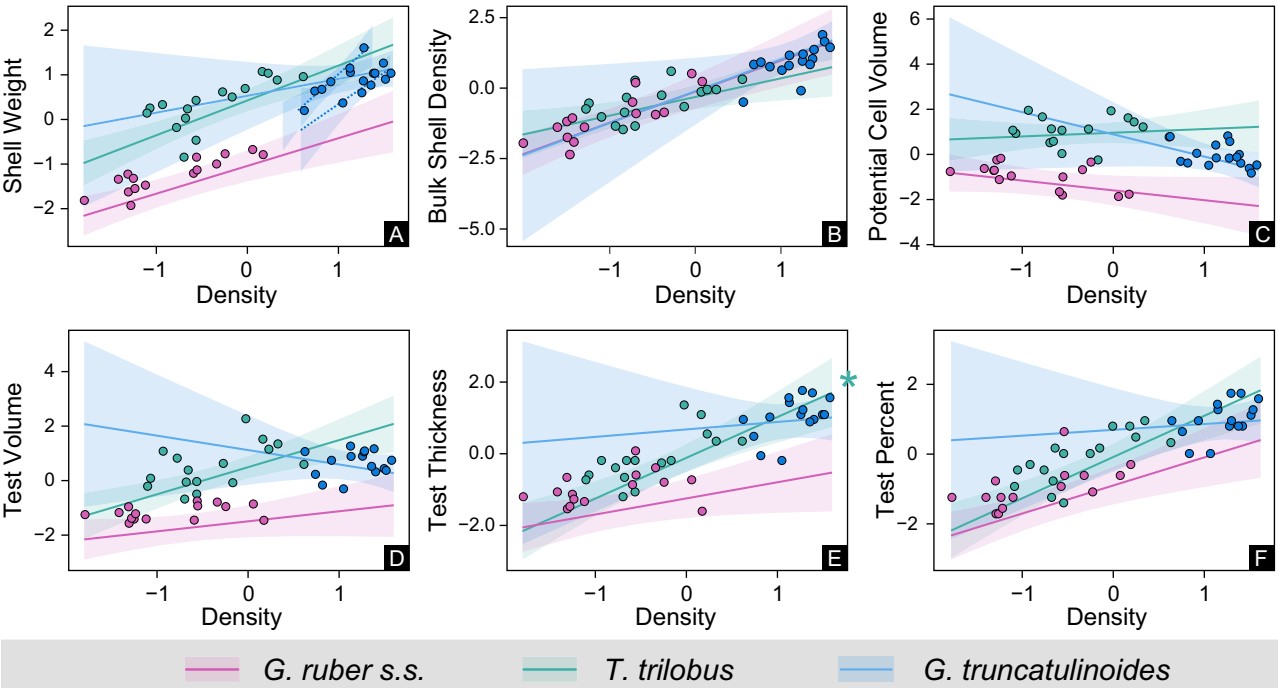

**Fig. 3 | Marginal effects plots showing the relationships between each of the six morphological characteristics and density, considering the three species individually and when accounting for variation in salinity and [CO$_3^{2-}$].** Plots **A–C** correspond to raw shell mass, normalized mass, and total volume, respectively. Plots **D–F** show CT-derived digital traits: **D** test volume, **E** test thickness, and **F** test percentage. The shaded areas lie between the 5% and 95% confidence intervals of the linear model coefficients. Asterisks denote the species with responses to density significantly different (i.e., stronger) from those of *G. ruber* (interaction effect $p < 0.05$). All values were scaled to unit variance to enable comparisons between the traits.

This suggests that BSD-based reconstructions could provide reliable insights into past ocean density gradients across species.

Ocean density had a significantly greater effect on the test thickness of *T. trilobus* ($p = 0.03$) than it did on the other two species (Fig. 3). In addition, the test volumes of *T. trilobus* increased with increasing density, whereas the test volume of *G. truncatulinoides* and *G. ruber* did not substantially vary with increasing density. Finally, although the "potential cell" volume was insignificantly related to density across and within the three species, each species individually presented different relationships, with *G. ruber* and (to a lesser extent) *G. truncatulinoides* showing a decrease in "cell" volume with increasing density and *T. trilobus* showing no relationship.

**Species specific calcification response to seawater density**

The relationships between surface ocean density, shell weights and bulk shell densities, as identified previously (Fig. 3A, B), are presented unscaled in Fig. 4 for each species individually, without accounting for the other covariates (salinity and [CO$_3^{2-}$]). Furthermore, the formulas derived from the regression analysis are given in each graph and may be used to convert species-specific shell weights and overall BSDs into ambient seawater densities. These strong correlations support the role of seawater density as a key driver of foraminiferal calcification, rather than carbonate chemistry alone.

Since buoyancy force depends on the volume of displaced fluid, total cell volume plays a crucial role in regulating depth positioning in plankton[22]. Larger cells displace more seawater, increasing buoyant force and reducing sinking tendency, even in heavily calcified species. This explains why *T. trilobus*, despite having shell weights nearly as heavy as *G. truncatulinoides* (Fig. 2A), remains in surface waters in the region[25]. Its greater total "cell" volume (Fig. 2C) offsets its calcification, allowing it to stay afloat at depths similar to *G. ruber*. When normalized to total volume (BSDs), *T. trilobus* exhibits density traits consistent with surface-dwelling forms, aligning more closely with *G. ruber* (Fig. 2B). This pattern is further supported by digital

tomographic data alone (Fig. 2E, F), reinforcing that *T. trilobus* functionally aligns with surface-dwelling foraminifera.

Planktonic foraminifera BSDs (volume-normalized shell weights) show a highly significant, species-independent correlation with seawater density (Fig. 4A), reinforcing their potential use as a density proxy. The *G. ruber* test mass shows strongest correlation to density among globigerinids, whereas *T. trilobus* shows a slightly weaker relationship, likely due to differential dissolution effects, which have been observed to impact *T. trilobus* more significantly than other species[26]. In these two groups, the heavier specimens are found in the saltier and thus denser subtropical gyre waters, whereas the lightest specimens carry the geochemical signal of the lighter equatorial waters.

The calcification data for *G. truncatulinoides* do not follow a latitudinal trend, as light specimens are present in both equatorial and subtropical areas. However, the variations in shell weights and water densities are better explained when two distinct population groups are considered, as has been previously suggested for these central Atlantic samples[27]. The average *G. truncatulinoides* shell weight remains stable at 31.2 µg across both groups but analysing them separately reveals key differences. The first group shows an exceptionally strong correlation ($r^2 = 0.95$), with weights varying between 27.6 and 35 µg. The second group also shows a strong correlation ($r^2 = 0.76$) but has less weight variability, ranging from 28.5 to 33.2 µg (Fig. 4D). This suggests that while seawater density is a dominant control, additional ecological or physiological factors may influence the calcification response within distinct *G. truncatulinoides* populations. The divergence between the two populations lies in the geochemically reconstructed seawater densities but diminishes when shell weights are normalized to the total specimen ("potential cell") volumes (Fig. 4A). This normalization reinforces BSD as a species-independent seawater density proxy, reducing interspecies variability.

Overall, the rate at which foraminifera species increase their shell weight per unit change in water density (i.e., the slope of the regression line)

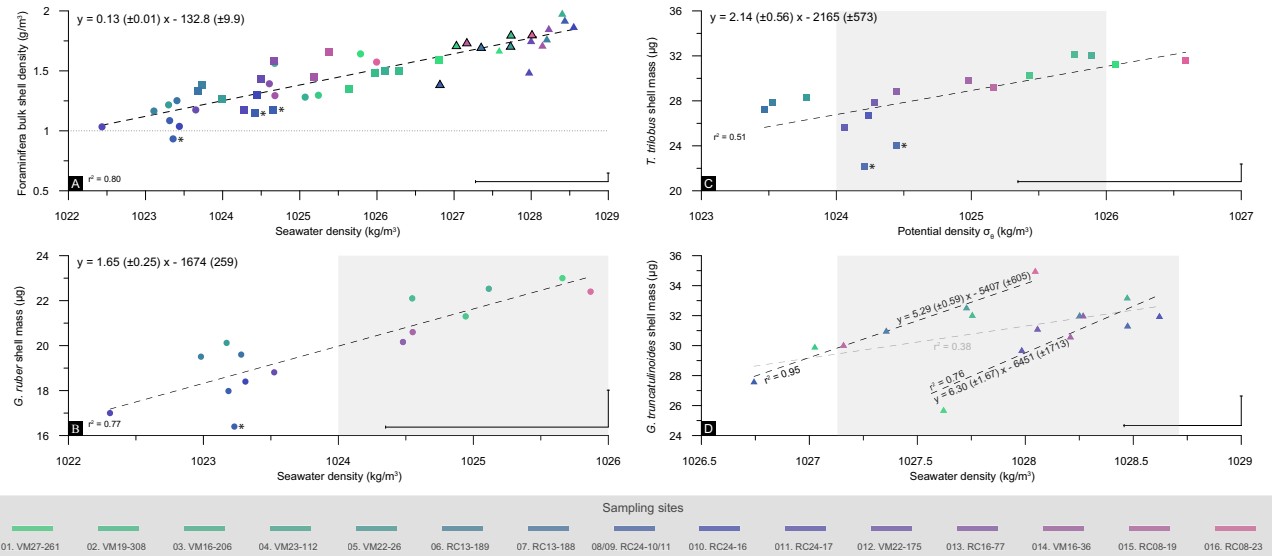

**Fig. 4 | Relationship between foraminiferal shell properties and seawater density. A** Correlation between bulk shell density (BSD) of planktonic foraminifera and ambient seawater density. Highlighted *G. truncatulinoides* data points denote a group; **B**–**D** species-specific shell weights with respect to seawater density. The data points are color-coded according to their site. Asterisks in the *G. ruber* and *T. trilobus* graphs mark samples of relatively enhanced dissolution. The gray shaded areas are $\sigma_\theta$ horizons from atlases at species-specific apparent calcification depths. Discrepancies between the atlases' $\sigma_\theta$ range and geochemically reconstructed densities may arise from "vital effects" on $\delta^{18}O_{shell}$ (see discussion). The $1\sigma$ confidence is depicted at the lower right of each graph.

is species-specific and appears to be dependent on habitat depth, increasing from the surface-dwelling *G. ruber* to the deeper-dwelling *G. truncatulinoides*. These findings further support the hypothesis that foraminiferal calcification serves as an adaptive buoyancy regulation mechanism, rather than being solely controlled by surface ocean carbonate chemistry.

## Discussion
### Planktonic foraminifera shell masses as indicators of seawater density

The present analysis revealed a general influence of seawater density on the calcification of planktonic foraminifera across all three species studied. Compared with previous work[9,28], in this study, we assess multispecies modern shell traits across ocean-wide oceanographic gradients, we do not find an overall dependency of the various shell traits on $[CO_3^{2-}]$. Instead, all the various shell traits related to calcification intensity, such as test percentage, weight, thickness, and BSD, respond primarily to seawater density, with the magnitude of this response varying among the species. In addition, the significant effects of density on "potential cell" volume, shell weight, and shell density, even when trait differences across species are considered, indicate that planktonic foraminifera may adapt to in situ oceanic environmental conditions. While traits involving shell weight measurements may be subject to detritus contamination[23], traits derived from 3D X-ray tomographic data analyses, such as test thickness and test percentage, are more precise species-specific indicators of calcification. The strong correlations between independently derived in situ oceanographic data and geochemical foraminifera shell information corroborate hydrography as a key driver of plankton calcification in the open ocean.

Planktonic foraminifera shell weights are valuable, easily measured biological indicators in geological records and are shown here to respond to seawater density changes at both the intra- and interspecies levels. The shell weights (Fig. 2A) and volume-normalized shell weights (Fig. 2B) of the different species increased with habitat depth despite the increase in oceanic acidity, reflecting an adaptation to increasing seawater density at greater depths. Similar trends can been observed in coccolithophores, which increase their cell density with depth by raising their particulate inorganic carbon to particulate organic carbon (PIC:POC) ratios (Supplementary Fig. 2). More specifically, the rate at which each species increased its shell weight in response to changes in ambient seawater density (regression

slopes; Fig. 4) potentially aligning with the inclination of the pycnocline, with more gradual slopes for the surface mixed-layer species[25] and steeper slopes for *G. truncatulinoides* at the base of the pycnocline[27]. Discrepancies between geochemically reconstructed (from combined Mg/Ca and $\delta^{18}O$) and in situ densities (Fig. 4) may stem from species-specific "vital effects" that are known to alter the $\delta^{18}O$ in foraminifera carbonate, possibly because of the photosynthetic activity of algal symbionts[29]. Indeed, discrepancies are greater for symbiont-bearing globigerinids[30] and smaller for the symbiont-barren *G. truncatulinoides*. Since the magnitude of these effects remains uncertain[31], no corrections were applied here for consistency.

The species-specific shell weight to seawater density relationships reported here (Fig. 4B, C) and elsewhere for upwelling species[18] may be simple yet powerful tools for ocean density reconstructions. However, if the slopes of these regression lines follow the structure of the pycnocline, they may vary spatiotemporally (e.g., ref. 32). Alternatively, of particular importance is the strong correlation between foraminifera shell weights normalized to their μCT-derived total volumes (BSDs), and seawater density (Fig. 4A). This strong relationship suggests that BSDs can be used as species-independent seawater density proxies to describe the surface and subsurface structure of the ocean. The BSD-density relationship converged more strongly across species than the other traits did (Fig. 3), which implies that size control may be another strategy used by nonmotile planktonic organisms for depth determination along water density gradients. However, since some species exhibit weak relationships between total "cell" size and seawater density, may indicate that size alone is not the primary factor in buoyancy regulation.

For *G. truncatulinoides*, the shell weight–density relationship (Fig. 4D) is more complex, aligning with distinct population groups in the central Atlantic[27,33]. The presence of *G. truncatulinoides* subspecies has been previously suggested[34,35]. Although the average shell weight is consistent across both variants, they yield slightly different ambient seawater densities. Since this discrepancy disappears when BSDs are considered, it implies that specimens in the different variants have different volumes. Indeed, the individuals from the two groups differ by ~5% in size (Table 2), despite both being picked from a size-restricted sieve fraction (300-355 μm). This sensitivity suggests that shell weight is a characteristic trait even among intraspecific species variants, indicating that planktonic forms may very precisely regulate shell weight.

**Table 2 | Summary table listing all the shell traits calculated for the different studied planktonic foraminifera species, together with the associated geochemical and in situ oceanographic data**

| Sample Location | Species | Shell weight (µg) | Cell Volume (nL) | Test Volume (nL) | Test % | Test thickness (µm) | BSD (g/cm³) | $\delta^{18}O_{shell}$ (‰) | Mg/Ca (mmol/mol) | $Temperature_{Mg/Ca}$ (°C) | $Depth_{Mg/Ca}$ (m) | Salinity at ACD | $CO_3^{2-}$ (µmol/kg) | Alkalinity (µmol/kg) | $Density_{Mg/Ca}$ (kg/m³) | 3D Salinity | 3D Density | 3D Temp. (°C) |
|---|---|---|---|---|---|---|---|---|---|---|---|---|---|---|---|---|---|---|
| 1. VM27-261 | G. ruber | 23.0 ± 7% | 14,011 ± 7% | 6,054 ± 19% | 37% | 6.7 | 1.64 | 0.01 | 3.597 | 23.4 | 50.8 | 36.9 | 249.7 | 2,395 | 1025.8 | 36.57 | 26.01 | 20.0 |
| 2. VM19-308 | G. ruber | 21.3 ± 8% | 16,630 ± 8% | 6,17 ± 18% | 37% | 7.2 | 1.28 | −0.66 | 4.386 | 25.5 | 38.3 | 37.2 | 287.1 | 2,431 | 1025.1 | 37.02 | 25.91 | 22.2 |
| 3. VM16-206 | G. ruber | 22.1 ± 6% | 14,150 ± 6% | 6,238 ± 19% | 45% | 7.9 | 1.56 | −0.93 | 4.739 | 26.3 | 59.4 | 37.4 | 291.9 | 2,440 | 1024.7 | 37.37 | 25.39 | 24.2 |
| 4. VM23-112 | G. ruber | 22.5 ± 10% | 17,372 ± 10% | 5,897 ± 11% | 34% | 6.6 | 1.30 | −0.59 | 4.417 | 25.6 | 67.1 | 37.1 | 279.3 | 2,399 | 1025.2 | 36.74 | 24.58 | 25.2 |
| 5. VM22-26 | G. ruber | 20.1 ± 9% | 16,547 ± 9% | 5,886 ± 16% | 36% | 6.8 | 1.22 | −1.25 | 3.810 | 24.0 | 65.6 | 36.2 | 285.9 | 2,360 | 1023.3 | 36.06 | 24.77 | 21.9 |
| 6. RC13-189 | G. ruber | 19.5 ± 8% | 16,741 ± 8% | 5,532 ± 11% | 33% | 6.2 | 1.17 | −1.40 | 3.813 | 24.0 | 70.0 | 36.0 | 242.2 | 2,364 | 1023.1 | 36.01 | 24.54 | 23.3 |
| 7. RC13-188 | G. ruber | 19.6 ± 6% | 15,663 ± 6% | 5,139 ± 17% | 33% | 6.1 | 1.25 | −1.54 | 4.425 | 25.6 | 64.1 | 36.1 | 243.2 | 2,373 | 1023.4 | 36.09 | 24.45 | 23.7 |
| 8. RC24-10 | G. ruber | 18.0 ± 8% | 16,576 ± 8% | 4,884 ± 17% | 30% | 5.5 | 1.08 | −1.33 | 3.376 (3.348) | 22.7 | 41.0 | 35.9 | 242.4 | 2,347 | 1023.3 | 35.90 | 24.13 | 24.1 |
| 9. RC24-11 | G. ruber | 16.4 ± 9% | 17,579 ± 9% | 5,186 ± 19% | 30% | 5.6 | 0.93 | −1.54 | 3.725 | 23.7 | 36.0 | 35.9 | 247.0 | 2,347 | 1023.4 | 35.91 | 24.07 | 24.4 |
| 10. RC24-16 | G. ruber | 18.4 ± 8% | 17,739 ± 8% | 5,458 ± 14% | 31% | 5.9 | 1.04 | −1.51 | 3.807 | 24.0 | 55.3 | 36.0 | 260.7 | 2,366 | 1023.4 | 35.89 | 24.34 | 23.1 |
| 11. RC24-17 | G. ruber | 17.0 ± 6% | 16,449 ± 6% | 5,409 ± 16% | 33% | 6.0 | 1.03 | −2.13 | 3.737 | 23.8 | 59.2 | 36.1 | 258.3 | 2,365 | 1022.4 | 35.87 | 24.42 | 22.8 |
| 12. V22-175 | G. ruber | 18.8 ± 7% | 16,026 ± 7% | 5,144 ± 10% | 33% | 5.8 | 1.17 | −1.58 | 4.122 | 24.8 | 59.9 | 36.4 | 255.3 | 2,401 | 1023.7 | 36.53 | 24.35 | 24.8 |
| 13. RC16-77 | G. ruber | 20.2 ± 11% | 14,476 ± 11% | 5,073 ± 14% | 35% | 6.5 | 1.39 | −0.97 | 3.732 | 23.8 | 53.7 | 36.8 | 260.2 | 2,402 | 1024.6 | 36.69 | 24.91 | 24.6 |
| 14. VM16-36 | G. ruber | 20.6 ± 9% | 15,923 ± 9% | 5,937 ± 18% | 37% | 6.9 | 1.29 | −0.77 | 3.542 | 23.2 | 60.9 | 36.5 | 261.2 | 2,397 | 1024.7 | 36.51 | 24.87 | 22.9 |
| 15. RC8-19 | G. ruber | 21.0 ± 6% | 15,533 ± 6% | 5,571 ± 15% | 40% | 7.6 | 1.35 | - | 3.956 | 24.4 | 43.6 | 36.6 | 259.6 | 2,381 | - | 36.38 | 24.79 | 23.8 |
| 16. RC8-23 | G. ruber | 22.4 ± 8% | 14,236 ± 8% | 5,064 ± 19% | 39% | 5.4 | 1.57 | 0.19 | 3.479 | 23.0 | 51.8 | 36.5 | 259.6 | 2,394 | 1026.0 | 36.15 | 24.98 | 22.5 |
| 1. VM27-261 | T. trilobus | 31.2 ± 8% | 20,807 ± 8% | 9,705 ± 19% | 47% | 8.3 | 1.50 | 0.01 | 3.619 | 24.0 | 44.3 | 36.9 | 250.3 | 2,394 | 1026.3 | 36.56 | 25.98 | 20.2 |
| 2. VM19-308 | T. trilobus | 30.2 ± 7% | 22,374 ± 7% | 11,231 ± 21% | 46% | 9.8 | 1.35 | −0.43 | 3.800 | 24.6 | 47.4 | 37.1 | 285.4 | 2,431 | 1025.6 | 37.02 | 25.92 | 22.1 |
| 3. VM16-206 | T. trilobus | 32.2 ± 10% | 21,689 ± 10% | 9,991 ± 15% | 46% | 9.4 | 1.48 | −0.23 | 3.998 | 25.3 | 71.8 | 37.4 | 289.4 | 2,437 | 1026.0 | 37.34 | 25.48 | 23.8 |
| 4. VM23-112 | T. trilobus | 32.0 ± 9% | 21,292 ± 9% | 9,384 ± 8% | 44% | 8.6 | 1.50 | −0.11 | 3.640 | 24.0 | 103.0 | 37.3 | 269.2 | 2,412 | 1026.1 | 36.94 | 25.01 | 24.1 |
| 5. VM22-26 | T. trilobus | 28.3 ± 6% | 22,429 ± 6% | 9,259 ± 20% | 39% | 7.5 | 1.26 | −1.49 | 3.503 | 23.5 | 69.2 | 36.2 | 285.7 | 2,359 | 1024.0 | 36.05 | 24.83 | 21.6 |
| 6. RC13-189 | T. trilobus | 27.9 ± 5% | 20,171 ± 5% | 7,617 ± 24% | 38% | 6.9 | 1.38 | −1.71 | 3.990 | 25.3 | 61.6 | 36.0 | 250.1 | 2,365 | 1023.7 | 36.03 | 24.34 | 24.0 |
| 7. RC13-188 | T. trilobus | 27.3 ± 8% | 20,482 ± 8% | 7,132 ± 24% | 35% | 6.7 | 1.33 | −1.70 | 3.572 | 23.8 | 76.9 | 36.0 | 231.1 | 2,370 | 1023.7 | 36.04 | 24.79 | 22.4 |
| 8. RC24-10 | T. trilobus | 24.1 ± 11% | 20,480 ± 11% | 6,653 ± 21% | 32% | 6.2 | 1.18 | −0.94 | 2.923 (2.840) | 21.1 | 47.3 | 35.8 | 235.8 | 2,348 | 1024.7 | 35.90 | 24.22 | 23.8 |
| 9. RC24-11 | T. trilobus | 22.1 ± 9% | 19,264 ± 9% | 6,352 ± 18% | 33% | 6.0 | 1.15 | −1.10 | 2.921 | 21.1 | 47.4 | 35.8 | 235.8 | 2,348 | 1024.4 | 35.90 | 24.22 | 23.8 |
| 10. RC24-16 | T. trilobus | 26.7 ± 5% | 20,612 ± 5% | 7,386 ± 21% | 36% | 6.8 | 1.30 | −1.12 | 3.146 | 22.1 | 66.9 | 36.0 | 251.2 | 2,362 | 1024.5 | 35.85 | 24.58 | 22.0 |
| 11. RC24-17 | T. trilobus | 25.6 ± 5% | 21,786 ± 5% | 8,836 ± 15% | 38% | 7.2 | 1.18 | −1.25 | 3.190 | 22.3 | 67.9 | 36.0 | 250.4 | 2,362 | 1024.3 | 35.84 | 24.60 | 21.9 |
| 12. V22-175 | T. trilobus | 27.8 ± 8% | 19,399 ± 8% | 8,113 ± 15% | 42% | 7.5 | 1.43 | −1.19 | 3.822 | 24.7 | 61.6 | 36.4 | 254.6 | 2,400 | 1024.5 | 36.52 | 24.37 | 24.7 |
| 13. RC16-77 | T. trilobus | 28.8 ± 7% | 18,231 ± 3% | 7,418 ± 12% | 41% | 7.5 | 1.58 | −1.05 | 3.527 | 23.6 | 56.8 | 36.8 | 259.2 | 2,402 | 1024.7 | 36.69 | 24.94 | 24.5 |
| 14. VM16-36 | T. trilobus | 29.8 ± 6% | 20,595 ± 6% | 8,52 ± 16% | 41% | 7.4 | 1.45 | −0.66 | 3.252 (3.240) | 22.5 | 68.1 | 36.5 | 260.2 | 2,395 | 1025.2 | 36.49 | 24.95 | 22.6 |
| 15. RC8-19 | T. trilobus | 29.2 ± 5% | 17,584 ± 5% | 7,347 ± 18% | 42% | 7.5 | 1.66 | −0.57 | 3.509 | 23.5 | 51.9 | 36.6 | 258.4 | 2,384 | 1025.4 | 36.35 | 24.89 | 23.4 |
| 16. RC8-23 | T. trilobus | 31.6 ± 9% | 19,822 ± 9% | 8,459 ± 13% | 43% | 8.3 | 1.59 | 0.43 | 3.087 (3.597) | 21.8 | 63.8 | 36.3 | 252.2 | 2,380 | 1026.8 | 36.11 | 25.14 | 21.9 |
| 1. VM27-261 | G. truncatulinoides | 28.5 ± 8% | 17,060 ± 8% | 6,974 ± 17% | 41% | 7.5 | 1.67 | 2.59 | 2.594 (2.224) | 16.3 | 339.0 | 36.2 | 201.1 | 2,359 | 1027.6 | 36.04 | 26.84 | 14.9 |
| 2. VM19-308 | G. truncatulinoides† | 29.9 ± 9% | 17,450 ± 9% | 7,855 ± 30% | 45% | 8.5 | 1.72 | 2.22 | 2.218 (2.142) | 13.7 | 560.2 | 35.8 | 157.5 | 2,351 | 1027.0 | 35.75 | 27.06 | 12.8 |
| 3. VM16-206 | G. truncatulinoides† | 32.1 ± 7% | 17,778 ± 10% | 8,943 ± 37% | 49% | 9.9 | 1.80 | 2.02 | 2.020 | 12.1 | 600.0 | 35.6 | 150.3 | 2,337 | 1027.7 | 35.65 | 27.13 | 12.2 |
| 4. VM23-112 | G. truncatulinoides† | 33.2 ± 10% | 16,766 ± 5% | 8,202 ± 27% | 48% | 9.4 | 1.98 | 2.54 | 2.536 | 15.9 | 281.6 | 36.2 | 187.4 | 2,348 | 1028.4 | 35.83 | 26.83 | 13.8 |
| 5. VM22-26 | G. truncatulinoides† | 32.6 ± 5% | 19,037 ± 8% | 9,57 ± 21% | 50% | 10.1 | 1.71 | 1.69 | 1.688 | 9.1 | 426.3 | 34.7 | 93.3 | 2,304 | 1027.7 | 34.78 | 27.16 | 8.3 |
| 6. RC13-189 | G. truncatulinoides | 32.1 ± 8% | 18,122 ± 6% | 9,407 ± 13% | 52% | 10.3 | 1.77 | 2.26 | 2.261 | 14.0 | 158.0 | 35.4 | 160.9 | 2,332 | 1028.2 | 35.43 | 26.46 | 14.4 |
| 7. RC13-188 | G. truncatulinoides† | 31.01 ± 6% | 18,231 ± 6% | 8,707 ± 21% | 47% | 9.3 | 1.70 | 2.13 | 2.130 | 13.0 | 208.1 | 35.3 | 140.7 | 2,323 | 1027.4 | 35.17 | 26.76 | 12.3 |
| 8. RC24-10 | G. truncatulinoides | 27.6 ± 3% | 19,863 ± 5% | 9,237 ± 4% | 47% | 9.2 | 1.39 | 1.38 | 1.382 | 5.8 | 617.8 | 34.5 | 80.4 | 2,305 | 1026.8 | 34.52 | 27.34 | 5.8 |
| 9. RC24-11 | G. truncatulinoides | 31.3 ± 5% | 16,298 ± 9% | 8,095 ± 20% | 49% | 9.4 | 1.92 | 1.86 | 1.856 | 10.7 | 326.6 | 35.0 | 96.3 | 2,316 | 1028.4 | 34.92 | 27.00 | 9.8 |
| 10. RC24-16 | G. truncatulinoides | 32.0 ± 9% | 17,094 ± 8% | 8,709 ± 16% | 51% | 10.1 | 1.87 | 2.36 | 2.360 | 14.7 | 123.0 | 35.5 | 144.6 | 2,333 | 1028.6 | 35.47 | 26.14 | 15.9 |

**Table 2 (continued) | Summary table listing all the shell traits calculated for the different studied planktonic foraminifera species, together with the associated geochemical and in situ oceanographic data**

| Sample Location | Species | Shell weight (µg) | Cell Volume (nL) | Test Volume (nL) | Test % | Test thickness (µm) | BSD (g/cm³) | $\delta^{18}O_{shell}$ (‰) | Mg/Ca (mmol/mol) | Temperature$_{Mg/Ca}$ (°C) | Depth$_{Mg/Ca}$ (m) | Salinity at ACD | $CO_3^{2-}$ (µmol/kg) | Alkalinity (µmol/kg) | Density$_{Mg/Ca}$ (kg/m³) | 3D Salinity | 3D Density | 3D Temp. (°C) |
|---|---|---|---|---|---|---|---|---|---|---|---|---|---|---|---|---|---|---|
| 11. RC24-17 | G. truncatulinoides | 29.7 ± 8% | 19.932 ± 8% | 8.957 ± 17% | 46% | 9.4 | 1.49 | 1.77 | 1.767 | 9.9 | 300.7 | 34.9 | 94.1 | 2,308 | 1028.0 | 34.93 | 26.98 | 10.1 |
| 12. V22-175 | G. truncatulinoides | 31.1 ± 8% | 17.762 ± 7% | 9.219 ± 9% | 52% | 10.4 | 1.75 | 2.68 | 2.682 | 16.9 | 126.7 | 35.8 | 213.3 | 2,373 | 1028.0 | 35.78 | 25.93 | 19.5 |
| 13. RC16-77 | G. truncatulinoides | 32.0 ± 7% | 17.270 ± 5% | 8.009 ± 20% | 46% | 9.2 | 1.85 | 2.03 | 2.034 | 12.3 | 224.5 | 35.2 | 147.6 | 2,309 | 1028.2 | 35.28 | 26.59 | 12.9 |
| 14. VM16-36 | G. truncatulinoides | 30.6 ± 5% | 17.853 ± 6% | 8.34 ± 33% | 46% | 9.1 | 1.72 | 2.64 | 2.640 | 16.6 | 181.7 | 35.7 | 185.3 | 2,343 | 1028.1 | 35.55 | 26.25 | 16.5 |
| 15. RC8-19 | G. truncatulinoides† | 30.1 ± 6% | 17.283 ± 6% | 7.203 ± 35% | 41% | 7.7 | 1.74 | 2.04 | 2.044 (2.283) | 12.3 | 363.1 | 35.1 | 164.4 | 2,318 | 1027.2 | 35.04 | 26.69 | 11.8 |
| 16. RC8-23 | G. truncatulinoides† | 35.0 ± 6% | 19.380 ± 7% | 9.276 ± 26% | 47% | 9.6 | 1.81 | 2.15 | 2.147 | 13.2 | 317.8 | 35.2 | 174.5 | 2,316 | 1028.0 | 35.17 | 26.55 | 13.2 |

Temperatures were converted from the Mg/Ca data via the calibration equation of ref. [66]. Daggers (†) denote a distinct G. truncatulinoides group. BSD = Bulk shell density and ACD = calcification depth. Refer to the Methods for a complete description of the parameters. The data are available also at Zarkogiannis et al. [82].

Furthermore, *G. truncatulinoides* shows another slightly distinct calcification response to density than the globigerinids (Fig. 3E). Compared with those of the other two species, the heaviest and densest shells are not always found in denser subtropical gyres. This species is predominantly thermocline-associated[4,36], but its habitat depth range varies based on encrustation stage[37]. In subtropical waters, *G. truncatulinoides* calcification begins near the surface, with secondary crust deposition occurring at ~700 m. In the tropics, although secondary crusting occurs at shallower depths (~400 m), the calcification of the primary shell takes place below the mixed layer at ~200 m[38]. Since the geochemical signature of bulk shell analyses represents an integrated signal over the specimen's ontogenetic depth migration[39–41], subtropical *G. truncatulinoides* exhibit lower average Mg/Ca values (Table 2), skewing seawater density reconstructions overall.

Such discrepancies in depth habitats may arise because, in encrusting species like *G. truncatulinoides*, the gametogenic crust calcite generally has a lower Mg/Ca ratio than earlier ontogenetic calcite, regardless of the depth at which it forms[42–44]. To assess whether potential overestimation of $ACD_{Mg/Ca}$ in heavily encrusted specimens, which have inherently lower Mg/Ca ratios, might affect the robustness of our results, we repeated our statistical analyses using, for *G. truncatulinoides* alone, oceanographic parameters extracted at $\delta^{18}O$-based apparent calcification depths $ACD_{iso}$ (see Supplementary Note). We found that the main results of the study remain consistent (Supplementary Table 5). The only notable difference was that the relationship between test thickness and density became significantly stronger when using $ACD_{iso}$ of $ACD_{Mg/Ca}$ (Supplementary Table 2d).

Changes in the shell weight of planktonic foraminifera have also been associated with variations in seawater salinity[23,45]. In the present study, salinity was found to exert some influence on the calcification of the mixed layer species *G. ruber* and *T. trilobus* (Supplementary Table 2) but not on that of *G. truncatulinoides* or the present dataset as a whole (Table 1). Salinity is important in determining the depth of the surface mixed layer[46], and in this central Atlantic area, there is a subsurface salinity maximum between ~50 and 150 m[47,48], below which salinities gradually decrease. However, since planktonic foraminifera calcification follows habitat depth and continues to increase rather than decrease in species residing below the surface mixed layer (e.g., *G. truncatulinoides*; Fig. 2A, B), this suggests that it is driven by the seawater density gradient rather than the salinity profile. Consequently, since salinity and density co-vary with depth in the mixed layer, surface-dwelling species respond similarly to both parameters, whereas at greater depths (e.g., *G. truncatulinoides* calcification depths) where these two hydrographic parameters diverge, the effect of density becomes evident.

Foraminiferal shell weight has also often been observed to decrease with increasing sea surface temperature[49,50]. In this study, such an inverse relationship is weak or absent, particularly for *G. truncatulinoides*, and this holds for other calcification metrics such as test percentage and BSD (Supplementary Table 2). Across the full dataset, no consistent effect of temperature on shell weight was detected, although slight negative correlations with test percentage and BSD were observed (Table 1). Consistent with previous studies, we interpret these inverse trends to reflect the indirect influence of temperature via its control on seawater density, which in the mixed layer of this region is primarily temperature-driven[51]. This supports further the broader conclusion that it is the physical property of density rather than temperature alone that better explains patterns of calcification in planktonic foraminifera.

The present analysis demonstrates that variations in planktonic foraminifera shell weight strongly correlate with changes in ambient seawater density, supporting the idea that foraminifera, as nonmotile plankton, adjust their overall density through biomineralization to match that of the surrounding fluid at specific depths along the pycnocline, allowing them to occupy optimal depth habitats. In the absence such adjustment, displacement will occur until the densities of the organism and the surrounding medium equilibrate. This identified dependency of calcification on surface ocean density implies adaptations of calcifying plankton as a response to anthropogenic oceanic warming and freshening due to melting ice sheets.

Modest decreases in calcification can lead to pronounced increases in surface ocean alkalinity levels and deepening of the carbonate compensation depth (CCD)[52]. A decrease in pelagic calcification in response to the ongoing decrease in seawater density could lead to long-term alkalinity build-up in the surface ocean, which has the potential to partially mitigate atmospheric carbon emissions. Furthermore, the active regulation of calcification in planktonic organisms for buoyancy adjustment would require reduced calcification needs in the future ocean, alleviating some of the known pressure exerted by ongoing ocean acidification[11].

The above comprehensive analysis aimed to elucidate the relationships between environmental conditions and shell traits within calcifying plankton communities. Along the studied central Atlantic transect, we observed strong relationships between surface ocean density and the degree of biomineralization in three planktonic foraminifera species, which highlights the importance of water density in driving variation and adaptation in these species. Individuals inhabiting denser waters consistently presented heavier, denser, and thicker shells. This correlation persisted across all three species studied, despite their differing depth habitats and life histories, indicating that this effect arises directly from adaptation rather than ecology. Our models suggest that seawater density exerts a stronger influence on foraminifera calcification than other environmental factors such as carbonate ion concentrations, and that their shell weight, particularly bulk shell density (volume normalized shell weight), may serve as a proxy for seawater density. The adaptation of planktonic calcifiers to changes in ocean density may indeed demonstrate aspects of their resilience, while also impacting the surface ocean's capacity to absorb atmospheric carbon.

Given the significance of these findings and their potential implications for planktonic communities and the carbon cycle, we advocate expanding this type of analysis to a broader set of taxa, including species with less well-defined depth distributions, and incorporating water column–collected specimens to reduce uncertainties inherent in (geochemically) reconstructed seawater densities.

## Methods

The present study is based on planktonic foraminifera specimens from a meridional transect of 16 core-top sediment samples spanning from 31°N to 25°S and their locations are situated along either flank of the Mid-Atlantic Ridge (Fig. 1). This is a subset of the samples used in previous studies[26,27,53], where information about the Holocene age of the samples can be found. We focus on size-restricted (300–355 μm) shells of three planktonic foraminifera species: *Globigerinoides ruber albus* sensu stricto (commonly referred to as *G. ruber* white s.s.; marinespecies.org:taxname:1629739), *Trilobatus trilobus* (a morphotype of *T. sacculifer* characterized by a three-chambered final whorl and slit-like aperture; marinespecies.org:taxname:1027267), and *Globorotalia truncatulinoides* (marinespecies.org:taxname:113453).

All specimens used in this study were previously assessed for preservation using μCT scanning[23,26]. In the present study, the same μCT-scanned tests were geochemically analysed for paired stable oxygen isotope (δ[18]O) and Mg/Ca measurements. These geochemical data were used to calculate key physical oceanic variables such as temperature, salinity, density and species-specific apparent calcification depths. Based on these data, we estimated species-specific calcification depths, at which corresponding in situ physicochemical oceanographic parameters were obtained from oceanic atlases (as detailed below).

Species with known habitat preferences were selected to span both shallow (surface) and deeper (subthermocline) waters. Shell weight means were calculated from 50 specimens per sample[23], while 20 μCT-scanned individuals were used for morphometric analysis. From the scanned individuals 20 *G. ruber*, 14 *T. trilobus*, and 14 *G. truncatulinoides* specimens per location were crushed, homogenized, and then separated for isotope and trace element analyses. The number of specimens analyzed varied according to the species used; however, in all instances approximately 400 μg of carbonate was used for the geochemical analyses. The tests were gently crushed between two glass plates to ensure that all the chambers were opened, and

any visible large coarse grains were removed. Approximately 350 μg were transferred to acid-cleaned vials using a damp sable brush for elemental analyses and the remaining ~50 μg was used for stable isotope analyses. All the generated data are given in Table 1.

### Trace element analyses

For the Mg/Ca analyses the standard cleaning protocol for foraminifera[54] was used. The treatment took place in the metal free suite of the Department of Earth Sciences, University of Oxford. After dissolution, the samples were centrifuged for 5 min (5000 rpm) to exclude any remaining insoluble particles from the analyses. The samples were diluted with 2% nitric acid water before analysis with inductively coupled plasma mass spectrometry (ICP-MS; Perkin Elmer NexION 350D at the Department of Earth Sciences, University of Oxford. Instrumental precision of the ICP-MS was monitored every five samples by analysis of an in-house standard solution with a Mg/Ca of 2.93 mmol mol$^{-1}$ (long-term standard deviation of 0.026 mmol mol$^{-1}$ or 0.88%).

To enable interlaboratory comparison, the ECRM752-1 limestone standard, with a reported Mg/Ca ratio of 3.75 mmol mol$^{-1}$ (ref. 55), was also analysed ($n = 3$), yielding an average value of $3.85 \pm 0.027$ mmol mol$^{-1}$. Procedural blank values below zero were measured. Negative values can apparently occur due to the nonlinearity of the calibration at very low values. All the Mg/Ca values are given in mmol mol$^{-1}$. Replicate measurements of 7 samples revealed an average standard deviation of 0.11 mmol mol$^{-1}$ (Table 2). To monitor the cleaning efficacy, Al/Ca, Fe/Ca, and Mn/Ca were measured alongside Mg/Ca. None of these ratios showed covariance with Mg/Ca (Supplementary Fig. 3).

### Stable isotopes

Samples ranging from 41 to 75 μg, in mass were analysed isotopically for δ[13]C and δ[18]O at the Department of Earth Sciences, University of Oxford, using a Thermo Delta V Advantage gas source mass spectrometer, fitted with a Gas Bench II peripheral. Within the Gas Bench II sample block, the powdered samples, in 12 mL He-flushed exetainers (Labco), are reacted with 100% phosphoric acid at 50 °C. The relative $^{13}C/^{12}C$ values are reported in the conventional δ$^{13}$C (‰) notation, relative to the Vienna Pee Dee Belemnite (V-PDB), by assigning a value of $+1.95$‰ exactly to NBS-19. The relative $^{18}O/^{16}O$ values are reported in the conventional δ$^{18}$O (‰) notation, relative to V-PDB, on a normalised scale such that the δ$^{18}$O of NBS-19 is $-2.2$‰. Reproducibility of in-house marble standard NOCZ during these analytical runs was 0.03‰ for δ$^{13}$C and 0.045‰ for δ$^{18}$O ($1\sigma$, $n = 5$), with average δ$^{13}$C = $2.17 \pm 0.03$ ($1\sigma$, $n = 5$) and δ$^{18}$O = $-2.03 \pm 0.045$‰ ($1\sigma$, $n = 5$) within the combined uncertainty of the long-term average of NOCZ (δ$^{13}$C = $2.17 \pm 0.06$; δ$^{18}$O = $-1.90 \pm 0.09$‰; $1\sigma$; $n = 120$).

### Oceanographic data

The mean annual and monthly ocean temperature and salinity data from the period between January 2004 and January 2020 were extracted for each core site from the International Argo Ocean Monitoring Program[56]. The values are quality-controlled and refer to data from the first 1000 m of the water column. Instead of extrapolating single-point hydrographic data at the exact core coordinates, average surface (2.5 m) temperature and salinity values for each site were extracted from an area of $0.1 \times 0.1$ decimal degrees (~$10 \times 10$ km) around the site location.

Ocean carbonate system data were taken from GLODAPv1.1[57]. Given the late Holocene age of the core tops, we corrected the modern seawater data for the acidifying influence of anthropogenic $CO_2$, by subtracting the GLODAPv1.1 anthropogenic Dissolved Inorganic Carbon (DIC) estimates from the total DIC values to provide an estimate of preindustrial DIC. These values, along with paired alkalinity, nutrient, and hydrographic data, were used to make preindustrial $CO_2$ system determinations for GLODAP using CO2sys.m v1.1[58]. This dataset was then imported to Ocean Data View[59] and carbonate system variables obtained for each site at species-specific apparent calcification depth range over an area of 100 km around the sampling

locations using Ocean Data View's 3D estimation tool. For consistency further oceanographic parameters (3D Temperature, 3D Salinity, and 3D Density) were calculated for the same depth ranges.

### Estimation of temperatures and apparent calcification depths (ACDs)

In the literature, foraminifera ACDs are calculated via either one or a combination of isotope or Mg/Ca temperature ($T_{Mg/Ca}$) methods. Since "vital effects" strongly offset shell oxygen isotopes from equilibrium values, calcification depth estimates using $\delta^{18}O$ may be less accurate[60]. In the present study, $T_{Mg/Ca}$ reconstructions were used for ACD determination. For *G. truncatulinoides*, given the absence of symbionts and the influence of secondary crust on total shell Mg/Ca, we also calculated $ACD_{iso}$ (see Supplementary Note). The Mg/Ca reconstructed temperatures (calculated following the procedures described below) for each species were compared with the mean annual temperature profile for sites between 23°N and 23°S and mean temperatures of the three warmest months of the year north of 23°N and south of 23°S, similar to previous studies[61]. For these extratropical locations, the oceanographic parameters were averaged only for the three warmest months, since studies have shown that the largest flux of foraminifera dwelling above the thermocline occurs during the warmest months of the year at these latitudes[62,63]. The water depth where shell $T_{Mg/Ca}$ matched that of the in situ water temperature profile was considered to reflect the calcification depth ($ACD_{Mg/Ca}$) of the respective species at the site.

While non-thermal factors like carbonate chemistry and salinity can influence Mg/Ca in planktonic foraminifera, their effects are species-specific and secondary to temperature[64,65]. Given their uncertainty and the inconsistent taxon-specific responses that make universal corrections unreliable, no adjustments were applied, and Mg/Ca ratios were used solely as temperature indicators for calcification depth estimates.

There are many published Mg/Ca—temperature calibration equations for various planktonic foraminifera species. Temperature estimates using seven different equations were compared with the in situ SSTs at each site along the transect (Supplementary Fig. 4). The best-fitting equations, which most accurately describe the data and predict calcification depths comparable to those reported in the literature, were those from ref. 66, which were also derived from the same region, for *G. ruber albus* [Mg/Ca = 0.4 (±0.28) * exp (0.094 (± 0.01) * $T$] Eq. 1] and *T. trilobus* [Mg/Ca = 0.6 (±0.16) * exp (0.075 (±0.006) * $T$) Eq. 2]. For *G. truncatulinoides* the ref. 27 calibration equation for samples shallower than 4 km [Mg/Ca = 0.975 (±0.07) * exp (0.06 (±0.007) * $T$) Eq. 3] was used. This equation was produced from the superset of the present samples and provides good characterization of the present dataset.

### Geochemical determination of seawater density

The combination of Mg/Ca ratios of foraminiferal shell calcite with $\delta^{18}O_{shell}$ isotope measurements has been used to reconstruct ambient $\delta^{18}O_{seawater}$ values and, subsequently, water salinities through the commonly applied technique[67,68]. For the two globigerinid species, the $\delta^{18}O_{shell}$, without any correction for vital effects, was converted to $\delta^{18}O_{seawater}$ using the Mulitza et al.[69], equations. For *G. truncatulinoides*, which is assumed to calcify in equilibrium with seawater without vital effects[70,71], the general quadratic Kim and O'Neil[72] equation rewritten by ref. 73 was used to convert $\delta^{18}O_{shell}$ to $\delta^{18}O_{seawater}$. Subsequently the $\delta^{18}O_{seawater}$ was converted to salinity values using the global $\delta^{18}O_{seawater}$—salinity relationship of Broecker[74] for all species. The geochemically derived temperature and salinity values were then entered into the equation of state to calculate in situ and potential seawater densities[18,75] at average depths of 50 m for *G. ruber*, 60 m for *T. trilobus* and 300 m for *G. truncatulinoides*. These depths were obtained by averaging the Mg/Ca-derived depth estimates for each species along the transect. The two different definitions of density serve different purposes: in situ density is the correct density to use when calculating the weight of the fluid and is relevant to buoyancy considerations, whereas potential density is the appropriate variable to employ when inferring ocean dynamics and in this form is given in ocean atlases.

The error in the $\delta^{18}O$ of seawater ($\sigma_{\delta w}$) is a combination of the error in the measurement of $\delta^{18}O_{shells}$ ($\sigma_{\delta c} = 0.08‰$) and the error in the Mg/Ca-derived pelagic temperature ($\sigma_T = 0.194‰$): $\sigma_{\delta w} = \sqrt{\sigma_T^2 + \sigma_{\delta c}^2}$, consequently $\sigma_{\delta w} = 0.21‰$. The total error in the ice-volume–corrected $\delta^{18}O$ of seawater is $\sigma_{\delta w\text{-ice}} = \sqrt{\sigma_{\delta w}^2 + \sigma_{SL}^2} = 0.23‰$. By assuming a linear relationship between $\sigma_{\delta w\text{-ice}}$ and salinity[76], the partial differential equations of ref. 75 yield an error for salinity $\sigma_S = 0.63$ psu and for in situ density $\sigma_t = 1.72$ kg m$^{-3}$.

### Shell traits from μCT analyses

Microcomputed tomography (μCT) provides high-precision spatial information, enabling the study of the three-dimensional mass distribution of foraminiferal tests and the quantification of detrimental detrital infillings. Shell traits for each sample were calculated from a subset of the specimens collected, with shell weight being an average 50 specimens[23] and the μCT-based traits being the average of approximately 20 specimens. μCT data of ~20 specimens per species per site for *G. ruber* s.s. were obtained from Zarkogiannis et al.[77,78] and for *T. trilobus* from Zarkogiannis[79]. Various shell traits resulting from the spatial analyses of the existed tomographs include the "*potential cell*" volume, which represents the total volume, assuming that the cell was alive with all its chambers filled with protoplasm, and the "*potential*" outer surface area (Fig. 5). Volume-normalized foraminiferal shell weights were determined by dividing the average shell weights by the average "potential cell" volume (Fig. 5C) for each sample. The ratio of mass to volume is a measure of density and thus volume-normalized shell weights are termed *bulk shell density* (BSD)[23]. Furthermore, the total *test surface* area and *test volume* (i.e., the volume of the calcite test) were determined, and the ratio of these two parameters was used as an estimate of the average *test thickness*. Finally, the ratio of the test volume to the "potential cell" volume, when expressed as a percentage, provides an estimate of the total amount of biomineralized space per unit volume within the foraminiferal cell. This is referred to here as the test percentage (*test %*).

### Statistical analyses

The key environmental parameters considered for the present statistical analyses encompass a range of factors, such as: a) geochemically reconstructed seawater in situ densities (*Density_{Mg/Ca}* in Table 2); b) Argo salinity at ACD (*Salinity* in Table 2); c) 3D Density; d) 3D Salinity; and e) 3D Temperature (refer to *Oceanographic Data* section). In our approach, the key oceanographic parameters were derived using Mg/Ca-based calcification depths. To avoid statistical circularity, we excluded Mg/Ca-derived temperature from our linear mixed models. However, to ensure that the relationships observed between foraminiferal traits and oceanographic variables were not solely driven by temperature, we conducted the analyses using the independent 3D Temperature estimates. This allowed us to verify that the patterns identified were not artifacts of temperature-based derivation.

In addition to the in situ salinity, temperature and density, we used the concentration of carbonate ions as predictors of the traits. In particular, the study addresses carbonate system parameters, specifically e) Alkalinity; and f) $CO_3^{2-}$ concentrations ([$CO_3^{2-}$]) at ACD. Alkalinity (ALK), however, exhibited an extremely high correlation ($r = 0.963$) with Argo salinity, making it statistically impractical to separate their individual effects within the model; therefore, it was not further considered in the LMMs. Instead, we used [$CO_3^{2-}$] to address the effects of the carbonate system as it was less correlated with other predictors (see correlation's Table 1) and also is traditionally associated with shell calcification traits. Substituting ALK in place of [$CO_3^{2-}$] in our models did not alter the selected results based on AICc, due to the strong correlation between these parameters. This interchangeability confirms that our conclusions remain robust regardless of whether ALK or [$CO_3^{2-}$] is used, reinforcing the validity of our chosen model structure.

Once we had selected the predictor variables to use in our analyses, we compared the six traits ("*potential cell*" volume, *shell weight*, *bulk shell density*, *test volume*, *test thickness*, and *test percent*) among the three species examined. We used Kruskal-Wallis tests to account for non-normality and differences in the variances between species, and to correct for multiple comparisons ($n = 6$), we used the Benjamini and Yekuetli adjustment[80].

**Fig. 5 | Example of tomograph segmentation using computed tomography (CT) data visualization software. A** The foraminiferal test is segmented in yellow, while regions covered by external debris are segmented in red; **B** Internal chamber voids and debris are grouped together (blue) to represent the volume that would have been occupied by protoplasm; **C** Protoplasm and shell volumes are combined (cyan) to estimate the total potential "cell" volume occupied by a living foraminifer.

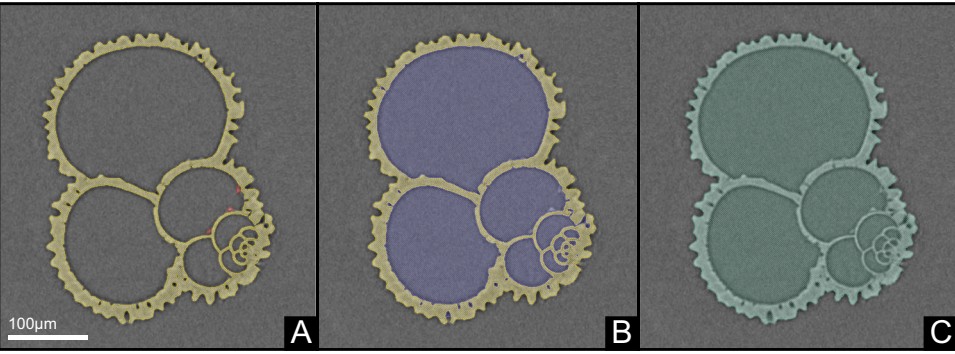

These adjustments minimize Type I error, providing a conservative estimate of the differences among species.

Because the six traits all varied significantly across species (see Fig. 2), we had to consider differences in the responses of each species to their surrounding environmental conditions. To achieve this goal, we employed two categories of linear models to analyse the relationships between each trait and the environmental variables while accounting for species identity. First, we developed linear mixed effects models (LMMs), treating species identity as a random. By incorporating species identity as a random intercept in these models, we controlled for the nonindependence of traits within each species (Fig. 2), thereby ensuring that this trait variation and the composition of species at each site did not obscure the relationship between environmental factors and traits. A significant result in these LMMs would indicate an association between traits and environmental variables irrespective of species identity.

We compared LMMs incorporating density (density model), salinity (salinity model), and both (density + salinity model), to a null model incorporating only carbonate ion concentrations $[CO_3^{2-}]$ using Akaike information criterion values corrected for small sample sizes (AICc), to determine which model best explained our data. For the LMM incorporating both density and salinity, we examined the effect sizes of the environmental predictors and their standard errors. The predictors used in these models are somewhat strongly correlated, leading to high variance inflation factors (VIF > 6) for the full model predicting shell density (Salinity = 7.1, $[CO_3^{2-}]$ = 10.7). However, this model was not selected for using AICc (see Table 1). All other models had VIF values less than 6, indicating that this multicollinearity had little effect on the model parameters.

After examining the trait-environment relationships, irrespective of species, we then reincorporated species identity as an intended, fixed effect to examine how density influences the traits of each species independently. For each trait, we used a model incorporating density and $CO_3^{2-}$ and converted species identity from a random effect to a fixed effect that interacted with density. These linear models explicitly model the response of each species to density. As with the linear mixed effects models, multicollinearity had little effect on the model parameters (all VIF < 4). Before generating both sets of linear models, we centred and scaled the trait and environmental data to unit variance so that the effect sizes of each model were comparable and informative. Each of the models in which effect sizes and confidence intervals were required (the density + salinity LMMs and all fixed effect models) were checked for heteroscedasticity and normality using the Breusch-Godfrey test in the R package "lmtest"[81]. For the fixed effect model using shell weight as the response, the model was significantly heteroscedastic ($p = 0.016$), likely due to the differences between the two populations of *G. truncatulinoides* (see above). Therefore, we modified this model using weighted regression (the significance and magnitude of the coefficients were only negligibly affected). All models were fit using log-likelihood.

All the statistical analyses were conducted in R (v. 4.3, R Core Team 2024), and a list of packages used and their versions can be found in Supplementary Table 3.

## Reporting summary
Further information on research design is available in the Nature Portfolio Reporting Summary linked to this article.

## Data availability
All data supporting the findings of this study, including those used to generate the figures are available at Figshare: https://doi.org/10.6084/m9.figshare.29459315.

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

## Acknowledgements
S.D.Z. would like to acknowledge the UK Research and Innovation Grant (SODIOM) EP/Y004221/1. B.R.S. is funded by the NSF EAR Postdoc Fellowship 2305234. J.W.B.R. was supported by European Research Council Horizon 2020 research and innovation programme grant agreement No. 805246. The authors would like to thank Dr Joost de Vries for producing Supplementary Fig. 2 from the CASCADE dataset and Dr Lennart de Nooijer for the valuable discussions. This research used samples provided by the Lamont-Doherty Core Repository.

## Author contributions
S.D.Z. conducted the laboratory analyses, contributed to formal analysis and data visualization, and led the generation of geochemical datasets. J.W.B.R. extracted and processed the in situ oceanographic data and calculated the carbonate system parameters. B.R.S. performed the statistical analyses and contributed to formal analysis and visualization. P.G.M. provided guidance on the geochemical measurements and contributed to data interpretation. All authors contributed to the writing and editing of the manuscript and approved the final version.

## Competing interests
The authors declare no competing interests.
