## [Transparent Peer Review file · Communications Earth & Environment]

Planktonic foraminifera regulate calcification according to ocean density

Corresponding Author: Dr Stergios Zarkogiannis

Version 0:

Decision Letter:

Dear Dr Zarkogiannis,

Please accept our apologies for the delay in sending a decision on your manuscript. Your manuscript titled "Floating, Sinking, and Sequestering: The role of ocean density in foraminiferal calcification" has now been seen by 2 reviewers, whose comments are appended below. You will see that they find your work of interest. However, they have raised concerns that must be addressed. In light of these comments, we cannot accept the manuscript for publication in its current form, but would be interested in considering a revised version that fully addresses these concerns.

Specifically, we suggest that you:

- fully demonstrate the role of ocean density in foraminiferal calcification
- consider the effects of water temperature on foraminiferal shell traits,
- rigorously describe your methodology.

When resubmitting, please provide a point-by-point response to the reviewers' comments. Please submit your responses as a separate file, distinct from your cover letter where you can add responses to the Editors' comments that you do not want to be made available to the reviewers. Word files are preferred. We recommend that any figures, tables or graphs that are included in the response to reviewers are also included in the main article or Supplementary Information.

If the revision process takes significantly longer than three months, we will be happy to reconsider your paper at a later date, as long as nothing similar has been accepted for publication at Communications Earth & Environment or published elsewhere in the meantime.

Please use the following link to submit your revised manuscript, point-by-point response to the reviewers' comments with a list of your changes to the manuscript text (which should be in a separate document to any cover letter), a tracked-changes version of the manuscript (as a PDF file) and any completed checklist:

Link Redacted

Please do not hesitate to contact us if you have any questions or would like to discuss the required revisions further. Thank you for the opportunity to review your work.

Best regards,

Yiming Wang, PhD

Editorial Board Member
Communications Earth & Environment
orcid.org/0000-0003-3228-5592

Alice Drinkwater, PhD
Associate Editor
Communications Earth & Environment
Consulting Editor
Communications Sustainability

EDITORIAL POLICIES AND FORMAT

If you decide to resubmit your paper, please ensure that your manuscript complies with our editorial policies and complete and upload the checklist below as a Related Manuscript file type with the revised article:

Editorial Policy Policy requirements
(Download the link to your computer as a PDF.)

- Behavioural and social science
- Ecological, evolutionary & environmental sciences
- Life sciences

<https://www.nature.com/documents/nr-reporting-summary.zip>

For your information, you can find some guidance regarding format requirements summarized on the following checklist: (<https://www.nature.com/documents/commsj-phys-style-formatting-checklist-article.pdf>) and formatting guide (<https://www.nature.com/documents/commsj-phys-style-formatting-guide-accept.pdf>).

REVIEWER COMMENTS:

Reviewer #1 (Remarks to the Author):

It has been a matter of great pleasure for me that I reviewed the manuscript entitled "Floating, Sinking, and Sequestering: The role of ocean density in foraminiferal calcification" by Stergios D. Zarkogiannis and others for possible publication in Communications Earth & Environment (COMMSJ-25-1114).

Zarkogiannis et al. conducted chemical analyses (Mg/Ca and $\delta^{18}O$) of three species of planktonic foraminifera in surface sediments in the central Atlantic, and compared shell traits (e.g., shell volume, weight, and density) with environmental factors (e.g., density, salinity, and $[CO_2]$). The results indicate that foraminiferal shell density seems to be a proxy for seawater density. This paper contains significant implications that should be shared with the marine environmental research community. In my opinion, the author's paper makes an important contribution to the research on foraminiferal calcification. If the following major and minor points are properly addressed and improved, this manuscript would make a worthy contribution to Communications Earth & Environment.

Major comments:

- 1) Seawater density is one of the most crucial environmental factors in this study. In general, seawater density is mainly regulated by changes in temperature rather than salinity. Furthermore, a great number of studies have shown that temperature is strongly related to the distribution, calcification and biology of foraminifera (e.g., Schiebel and Hemleben, 2017). In this paper, the authors examine the effects of density, salinity and $[CO_2]$ on foraminiferal traits (Table 1). However, they do not mention about temperature (I think, temperature is not mentioned anywhere in the article). Therefore, the author should consider and discuss the effects of water temperature on foraminiferal shell traits.
- 2) The three types used in this study are described in the manuscript as *Globigerinoides ruber albus*, *Trilobatus trilobus*, and *Globorotalia truncatulinoides* (L62-63). I think many readers may not be familiar with "*Globigerinoides ruber albus*" and "*Trilobatus trilobus*" because these two are not very common terms in the field of paleoceanography. So, please add a brief explanation such as "*G. ruber* s.s. (Fig. S3 and Table S2)", "*G. ruber* white s.s.", and "*T. sacculifer* with three-chamber and slit-type aperture".
- 3) "Bulk shell density" is one of the most important terms in this study. Many readers, including me, will think that the bulk shell density means to be calculated based on the actual volume of the shell, not the cell (i.e., assuming that the cell was alive with all its chambers filled with protoplasm). The authors explain in detail (L414-425), however, I think it is very significant term, so please add a brief explanation in the section except for "Materials and Method".

Minor comments:

L25-27: I think "will respond to the anthropogenic oceanic rarefaction, with potential implications for the carbon cycle" is a slightly difficult to understand, so please rephrase it.

L99-100: "the common assumption that carbonate ion availability is the dominant control on calcification" Please add the reference.

L106: "shell volume" is "cell volume"?

L113: "Ocean density had a significantly greater effect on the test thickness of *T. trilobus* than it did on the other two species ($p = 0.03$). In addition, the test volumes..." If the following sentences are based on Fig. 3, please refer it.

L114: "...the test volumes of "both" *T. trilobus*.... "both" is OK?

L134: "shell weights nearly as ...(Figure 2b)" is "2a"? Some of the shell trait graphs in Fig. 2 and Fig. 3 are in different positions (e.g., shell weights are "a" in Fig2 and "b" in Fig3), which may cause a simple mistake. I think it would be better to put them in the same place also for readers.

L134: "greater total volume (Figure 2a)" is "2d"?

L136: "normalized to total volume (BSDs), its flotation ... (Figure 2c)" is "2b"?

L154-155: "but has less weight variability, ranging from 28.5 to 33.2 μg (Figure 4d)." If you have a reason for excluding the triangular-green-sample around 26 in Figure 4d, please add it.

L188: "The shell weights (Figure 2b)" is "2a"? See below comment (L134).

L189: "volume-normalized shell weights (Figure 2c)" is "2b"? See below comment (L134).

L193: "This is similar to" Please delete it.

L195-197: "potentially aligning with the inclination of the pycnocline, with more gradual slopes for the surface mixed-layer species and steeper slopes for *G. truncatulinoidea* at the base of the pycnocline" Please add the reference about habitats for three species.

L197-198: "Discrepancies between in situ and geochemically reconstructed densities (from combined Mg/Ca and $\delta^{18}\text{O}$)" Sorry, I could not find these discrepancies in your manuscript (or in figures/tables?). So please explain these in more detail.

L225-226: "Furthermore, *G. truncatulinoidea* shows another slightly distinct calcification pattern with a significantly stronger response to density than the globigerinids (Figure 3e)." In Figure 3e, I do not recognize "slightly distinct calcification pattern with a significantly stronger response to density". Would you explain in more detail?

L244-245: "decrease in species residing below the surface mixed layer (e.g., *G. truncatulinoidea*; Figure 2b, c)" is "2a, b"? See below comment (L134).

L246-247: "since salinity and density co-vary with depth in the mixed layer" Since the density changes with temperature and salinity, please add a discussion of temperature as well. See below major comment #1.

Figure 2: It is hard to see A, B, etc. in the lower right corner of each graph, so could you make them a little bit easier to see (e.g., larger, bold)?

If the above major and minor points are adequately addressed and improved, I think this manuscript can make a fine contribution to *Communications Earth & Environment*. I hope my comments are helpful in improving their paper.

Reviewer #2 (Remarks to the Author):

The paper by Zarogiannis et al. presents an argument that calcification in planktonic foraminifera is associated with seawater density regardless of species. The implication is that calcification is a part of this group's buoyancy regulation function, and moreover that calcification may decrease with decreasing ocean density, providing a potentially important feedback to the carbon cycle.

Especially in light of studies such as Barrett et al., it is important to reevaluate potential mechanisms (if a singular one exists) for foraminiferal calcification. The main critique I have of this contribution as written is that I found it very difficult to understand the methods used. Even after several close reads, I am unable to find some key information that I believe should be included. For example: How many foraminifera were measured by CT? Of which species and at which sites? Were these

the same foraminifera measured for geochemistry, or just from the same general population? How was each trait calculated? While something like potential cell volume is relatively straight forward, this is not the case for all traits. For example, which volume measurement was the important BSD normalized to? One more related point is that it looks like the authors have done all statistics around means (though this is never explicitly stated). What does the variability of traits around those means actually look like in these populations?

One other area it would be interesting to see the authors explore is that of species-level traits (really BSD is what is discussed by the authors) and depth. The authors are using a 3-species succession in which the shallowest dwelling species is least dense compared to the deepest, but I think there are plenty of counter examples as well. For example, *G. scitula* and *M. cultrata* share a similar overall shape, with the later being both overall shallower and more heavily calcified. One could also consider *G. bulloides* and *G. ruber*, both spinose with comparable globular shapes, yet the less calcified *G. bulloides* is generally accepted as living deeper within the water column. Thus, the 3-species comparison here may make the inter-species comparisons seem a bit more clear cut than they are in actuality.

Finally, I have a comment that is applicable primarily to *truncatulinoides*. Note that the denser, crust calcite in *truncatulinoides* has lower Mg/Ca than earlier "ontogenetic" calcite. In most crusting species, this has been shown to be the case regardless of calcification depth (e.g., Davis et al., 2017; Reynolds et al., 2018; Hupp & Fehrenbacher, 2023). Thus, using only Mg/Ca as a depth indicator is likely to make groups with more crust (more heavily calcified) appear to have calcified deeper in the water column. I suggest given the expected correlation here, the use of Mg/Ca for depth habitat reconstruction (especially when that is then correlated against BSD) is likely not appropriate in species that form a gametogenic crust.

Minor points:

- Line 18: could the authors be more specific than "part of the surface carbon"?
- Line 30: bodies aren't enclosed in the shells – cytoplasm also flows outside of the shell
- Line 32: Sorry, this is a bit pedantic, but there are many active swimmers that are still generally considered planktonic rather than nektonic (e.g., copepods) in part due to their size and relatively small distances travelled despite active swimming behaviors
- Line 193: "This is similar to" is free floating.
- Please make the table 2 caption more descriptive? It is difficult to interpret several of these columns
- I would suggest adding a figure of surface density through the study area (and perhaps temperature and salinity?)

Citations:

Reynolds, C. E., Richey, J. N., Fehrenbacher, J. S., Rosenheim, B. E., & Spero, H. J. (2018). Environmental controls on the geochemistry of *Globorotalia truncatulinoides* in the Gulf of Mexico: Implications for paleoceanographic reconstructions. *Marine Micropaleontology*, 142, 92-104.

Davis, C. V., Fehrenbacher, J. S., Hill, T. M., Russell, A. D., & Spero, H. J. (2017). Relationships between temperature, pH, and crusting on Mg/Ca ratios in laboratory-grown *Neogloboquadrina* foraminifera. *Paleoceanography*, 32(11), 1137-1152.

Hupp, B. N., & Fehrenbacher, J. S. (2023). Geochemical differences between alive, uncrusted and dead, crusted shells of *Neogloboquadrina pachyderma*: Implications for paleoreconstruction. *Paleoceanography and Paleoclimatology*, 38(10), e2023PA004638.

Communications Earth & Environment is committed to improving transparency in authorship. As part of our efforts in this direction, we are now requesting that all authors identified as 'corresponding author' create and link their Open Researcher and Contributor Identifier (ORCID) with their account on the Manuscript Tracking System prior to acceptance. ORCID helps the scientific community achieve unambiguous attribution of all scholarly contributions. You can create and link your ORCID from the home page of the Manuscript Tracking System by clicking on 'Modify my Springer Nature account' and following the instructions in the link below. Please also inform all co-authors that they can add their ORCIDs to their accounts and that they must do so prior to acceptance.

If you experience problems in linking your ORCID, please contact the Platform Support Helpdesk.

Version 1:

Decision Letter:

Dear Dr Zarkogiannis,

Your manuscript titled "Floating, Sinking, and Sequestering: The role of ocean density in foraminiferal calcification" has now been seen by our reviewers, whose comments appear below. In light of their advice we are delighted to say that we are happy, in principle, to publish a suitably revised version in Communications Earth & Environment.

We therefore invite you to revise your paper one last time to address the remaining concerns of our reviewers. At the same time we ask that you edit your manuscript to comply with our format requirements and to maximise the accessibility and therefore the impact of your work.

EDITORIAL REQUESTS:

****Please take care to match our formatting and policy requirements. We will check revised manuscript and return manuscripts that do not comply. Such requests will lead to delays. ****

SUBMISSION INFORMATION:

OPEN ACCESS:

Communications Earth & Environment is a fully open access journal. Articles are made freely accessible on publication. For further information about article processing charges, open access funding, and advice and support from Nature Portfolio, please visit <https://www.nature.com/commsenv/open-access>

Link Redacted

Best regards,

Alice Drinkwater, PhD
Associate Editor
Communications Earth & Environment
Consulting Editor
Communications Sustainability

REVIEWERS' COMMENTS:

Reviewer #1 (Remarks to the Author):

The manuscript has been well revised. However, could you please revise Figure 4 more clearly visible? It is difficult to see the letters, numbers, and the "C" and "D" in the lower left corner of each graph. If the above minor point is adequately addressed, I think this manuscript will be acceptable for publication in Communications Earth & Environment.

Reviewer #2 (Remarks to the Author):

I am satisfied that all of my comments have been fully addressed. I will add just two minor suggestions around language.

Line 65: deeper -> subthermocline depths (or other more descriptive term)

Line 72: "total virtual "cell" volume" -> internal shell volume

** Visit Nature Portfolio's author and referees' website at www.nature.com/authors for information about policies, services and author benefits**

Reviewer #1 (Remarks to the Author):

It has been a matter of great pleasure for me that I reviewed the manuscript entitled "Floating, Sinking, and Sequestering: The role of ocean density in foraminiferal calcification" by Stergios D. Zarkogiannis and others for possible publication in *Communications Earth & Environment* (COMSENV-25-1114).

Firstly, we want to thank the reviewer for their time and effort to review our work and we are delighted that the reviewer shares the same views about the importance of our contribution.

Zarkogiannis et al. conducted chemical analyses (Mg/Ca and $\delta^{18}\text{O}$) of three species of planktonic foraminifera in surface sediments in the central Atlantic, and compared shell traits (e.g., shell volume, weight, and density) with environmental factors (e.g., density, salinity, and $[\text{CO}_2]$). The results indicate that foraminiferal shell density seems to be a proxy for seawater density. This paper contains significant implications that should be shared with the marine environmental research community. In my opinion, the author's paper makes an important contribution to the research on foraminiferal calcification. If the following major and minor points are properly addressed and improved, this manuscript would make a worthy contribution to *Communications Earth & Environment*.

Major comments:

1) Seawater density is one of the most crucial environmental factors in this study. In general, seawater density is mainly regulated by changes in temperature rather than salinity. Furthermore, a great number of studies have shown that temperature is strongly related to the distribution, calcification and biology of foraminifera (e.g., Schiebel and Hemleben, 2017). In this paper, the authors examine the effects of density, salinity and $[\text{CO}_2]$ on foraminiferal traits (Table 1). However, they do not mention about temperature (I think, temperature is not mentioned anywhere in the article). Therefore, the author should consider and discuss the effects of water temperature on foraminiferal shell traits.

It is indeed true that we have not addressed in the final text the effect of temperature although we did consider it during the study. The reason for this was to avoid comments on circularity in thinking since the extraction of all oceanographic parameters considered in the study were based on temperature-based depth estimates. We did however consider spatially averaged temperature from the world ocean atlas, which are independent estimates, and we now include these analyses in the revised version of the manuscript as well.

2) The three types used in this study are described in the manuscript as *Globigerinoides ruber albus*, *Trilobatus trilobus*, and *Globorotalia truncatulinoides* (L62-63). I think many readers may not be familiar with "*Globigerinoides ruber albus*" and "*Trilobatus trilobus*" because these two are not very common terms in the field of paleoceanography. So, please add a brief explanation such as "*G. ruber* s.s. (Fig. S3 and Table S2)", "*G. ruber white s.s.*", and "*T. sacculifer* with three-chamber and slit-type aperture".

Thank you for your helpful suggestion. We have clarified the taxonomic descriptions of *Globigerinoides ruber albus* and *Trilobatus trilobus* by adding commonly used terms and brief morphological notes as suggested. We have additionally added marinespecies.org identifiers. Figure S3 and Table S2 were corrected. These changes have been implemented in the revised manuscript.

3) "Bulk shell density" is one of the most important terms in this study. Many readers, including me, will think that the bulk shell density means to be calculated based on the actual volume of the shell, not the cell (i.e., assuming that the cell was alive with all its chambers filled with protoplasm). The authors explain in detail (L414-,425), however, I think it is very significant term, so please add a brief explanation in the section except for "Materials and Method".

We understand this concern. We have now added a brief explanation of bulk shell density (BSD) in the Introduction to clarify its meaning and how it is calculated, as requested (Lines 71-73). We furthermore introduced an additional figure (Figure 5) to depict the notion of "Potential Cell Volume".

Minor comments:

L25-27: I think "will respond to the anthropogenic oceanic rarefication, with potential implications for the carbon cycle" is a slightly difficult to understand, so please rephrase it.

We have rephrased the sentence for clarity in the revised manuscript. We retained, in parentheses, the term "rarefication" as a concise, single-word antonym to "densification," used here to describe anthropogenically driven reductions in ocean density.

L99-100: "the common assumption that carbonate ion availability is the dominant control on calcification" Please add the reference.

Reference added.

L106: "shell volume" is "cell volume"?

We indeed meant "cell volume". Thank you for spotting that. We have corrected similar inconsistencies throughout the text.

L113: "Ocean density had a significantly greater effect on the test thickness of *T. trilobus* than it did on the other two species ($p = 0.03$). In addition, the test volumes..." If the following sentences are based on Fig. 3, please refer it.

Reference to Figure 3 added.

L114: ...the test volumes of "both" *T. trilobus*.... "both" is OK?

Both was a mistake. Corrected, thank you.

L134: "shell weights nearly as ...(Figure 2b)" is "2a"? Some of the shell trait graphs in Fig. 2 and Fig. 3 are in different positions (e.g., shell weights are "a" in Fig2 and "b" in Fig3), which may cause a simple mistake. I think it would be better to put them in the same place also for readers.

The figures have been reorganized as suggested to ensure consistency in the placement of shell trait graphs across Figures 2 and 3. All corresponding references in the text have also been updated accordingly.

L134: "greater total volume (Figure 2a)" is "2d"?

Corrected

L136: "normalized to total volume (BSDs), its flotation ... (Figure 2c)" is "2b"?

Corrected

L154-155: "but has less weight variability, ranging from 28.5 to 33.2 μg (Figure 4d)." If you have a reason for excluding the triangular-green-sample around 26 in Figure 4d, please add it.

L188: "The shell weights (Figure 2b)" is "2a"? See below comment (L134).

Corrected

L189: "volume-normalized shell weights (Figure 2c)" is "2b"? See below comment (L134).

Corrected

L193: "This is similar to" Please delete it.

Deleted

L195-197: "potentially aligning with the inclination of the pycnocline, with more gradual slopes for the surface mixed-layer species and steeper slopes for *G. truncatulinoides* at the base of the pycnocline" Please add the reference about habitats for three species.

References added

L197-198: "Discrepancies between in situ and geochemically reconstructed densities (from combined Mg/Ca and $\delta^{18}\text{O}$)" Sorry, I could not find these discrepancies in your manuscript (or in figures/tables?). So please explain these in more detail.

The discrepancies are now better highlighted in the caption of Figure 4.

L225-226: "Furthermore, *G. truncatulinoides* shows another slightly distinct calcification pattern with a significantly stronger response to density than the globigerinids (Figure 3e)." In Figure 3e, I do not recognize "slightly distinct calcification pattern with a significantly stronger response to density". Would you explain in more detail?

We apologise for the confusion—'significantly stronger' was incorrect and should, in fact, have stated the opposite. However, as this point is already addressed in the paragraph between lines 153–167, we have removed any reference to it from this section.

L244-245: "decrease in species residing below the surface mixed layer (e.g., *G. truncatulinoides*; Figure 2b, c)" is "2a, b"? See below comment (L134).

Corrected

L246-247: "since salinity and density co-vary with depth in the mixed layer" Since the density changes with temperature and salinity, please add a discussion of temperature as well. See below major comment #1.

A discussion has been added. Please see lines 266–275. We thank the reviewer for the opportunity to address this point, as the relationship with density provides a clear explanation for the trends in shell weights previously reported for the mixed layer.

Figure 2: It is hard to see A, B, etc. in the lower right corner of each graph, so could you make them a little bit easier to see (e.g., larger, bold)?

The panel labels have been enlarged to improve visibility, as recommended.

If the above major and minor points are adequately addressed and improved, I think this manuscript can make a fine contribution to Communications Earth & Environment. I hope my comments are helpful in improving their paper.

We sincerely thank you for your thorough and constructive review. Your comments have been highly valuable and have significantly contributed to improving the quality of the manuscript.

Reviewer #2 (Remarks to the Author):

The paper by Zarkogiannis et al. presents an argument that calcification in planktonic foraminifera is associated with seawater density regardless of species. The implication is that calcification is a part of this group's buoyancy regulation function, and moreover that calcification may decrease with decreasing ocean density, providing a potentially important feedback to the carbon cycle.

Especially in light of studies such as Barrett et al., it is important to reevaluate potential mechanisms (if a singular one exists) for foraminiferal calcification. The main critique I have of this contribution as written is that I found it very difficult to understand the methods used. Even after several close reads, I am unable to find some key information that I believe should be included. For example: How many foraminifera were measured by CT? Of which species and at which sites? Were these the same foraminifera measured for geochemistry, or just from the same general population? How was each trait calculated? While something like potential cell volume is relatively straight forward, this is not the case for all traits. For example, which volume measurement was the important BSD normalized to? One more related point is that it looks like the authors have done all statistics around means (though this is never explicitly stated). What does the variability of traits around those means actually look like in these populations?

We agree with the reviewer that this contribution is timely, particularly considering the recent study by Barrett et al. While Barrett et al. mention seawater density as a potential driver of foraminiferal calcification, they do not explicitly test this hypothesis. We believe our study provides a timely and complementary advance by directly examining this mechanism.

We thank the reviewer for raising these important points regarding the clarity of our methods. In response, we have substantially revised the Methods section to clearly state the number of foraminifera measured by μ CT per species and per site, and we have clarified that the same specimens previously scanned were used for geochemical analyses. We now explicitly detail how each morphological trait was calculated, including the reference volume used for bulk shell density (BSD) specifically, the total apparent "cell" volume derived from μ CT data by including a new Figure (5). We have now provided a more transparent description of the variability around these means, as illustrated in the updated figure captions. These changes aim to improve reproducibility and interpretability of our approach.

The variability of traits around those means is now included in Figure 4. Additionally, we have revised the Table 2 to include all the uncertainties of the calculated shell traits.

One other area it would be interesting to see the authors explore is that of species-level traits (really BSD is what is discussed by the authors) and depth. The authors are using a 3-species succession in which the shallowest dwelling species is least dense compared to the deepest, but I think there are plenty of counter examples as well. For example, *G. scitula* and *M. cultrata* share a similar overall shape, with the later being both overall shallower and more heavily calcified. One could also consider *G. bulloides* and *G. ruber*, both spinose with comparable globular shapes, yet the less calcified *G. bulloides* is generally accepted as living deeper within the water column. Thus, the 3-species comparison here may make the inter-species comparisons seem a bit more clear cut than they are in actuality.

We appreciate this thoughtful observation. Our study was designed as a first-order test of the buoyancy-density hypothesis, and we therefore deliberately selected three species (*G. ruber*, *T.*

trilobus, *G. truncatulinoides*) with well-resolved and distinct depth habitats in our study area that were also consistently abundant (more than 50 shells) across all samples.

To date, there are no published shell-weight datasets or μ CT reconstructions for *G. scitula* or *M. cultrata* that would allow for reliable estimates of bulk shell density (BSD). Moreover, it is not always straightforward to determine which of these two species is more heavily calcified. *M. cultrata* often displays thinner, more transparent chamber walls that dissolve readily. The low thickness of the chamber walls may account for the frequent preservation of only the keel of this species in sedimentary records.

We agree with the point regarding *G. bulloides* and *G. ruber*. However, *G. bulloides* was not sufficiently abundant in all our samples and was therefore not included in the current analysis. Comparisons between these species must be made using specimens from the same sieve fraction, as *G. ruber* is typically slightly larger than *G. bulloides* and thus when normalizing to cell volume, *G. ruber* may exhibit lower BSD values. Even if not, physiological differences may still play a role. At present, key uncertainties remain: (i) which planktonic foraminifera osmoregulate and to what extent, since osmoregulation severely affects cell density, (ii) how symbiont activity modifies the composition of the protoplasm and thereby cell density, and (iii) what proportion of chambers remain filled with cytoplasm throughout most of the life cycle. Non-symbiotic taxa may inhabit only the inner chambers, using empty outer chambers as ballast or post-feeding space, whereas symbiotic species may retain cytoplasm in more chambers, thereby increasing surface area for symbionts. These differences can influence overall foraminiferal density of live cells, while BSD is just a normalization of the fossil shell weight assuming only that all chambers are filled.

We fully agree, however, that a broader, multi-species assessment is needed. A follow-up study including additional spinose and non-spinose, symbiotic and non-symbiotic species is already underway. We hope that the present manuscript will help stimulate further work along these lines. Future efforts will also incorporate newly acquired plankton tow samples.

We have added a few extra lines in the revised text discussing the comment raised.

Finally, I have a comment that is applicable primarily to *truncatulanoides*. Note that the denser, crust calcite in *truncatulanoides* has lower Mg/Ca than earlier "ontogenetic" calcite. In most crusting species, this has been shown to be the case regardless of calcification depth (e.g., Davis et al., 2017; Reynolds et al., 2018; Hupp & Fehrenbacher, 2023). Thus, using only Mg/Ca as a depth indicator is likely to make groups with more crust (more heavily calcified) appear to have calcified deeper in the water column. I suggest given the expected correlation here, the use of Mg/Ca for depth habitat reconstruction (especially when that is then correlated against BSD) is likely not appropriate in species that form a gametogenic crust.

We had initially calculated calcification depths using both the species-specific oxygen isotopic and Mg/Ca values of the shells. For the globigerinids the calcification depths generally agreed but not at the gyres, where we found large deviations between $ACD_{Mg/Ca}$ and ACD_{iso} . We think this may be because of vital effects due to symbionts in this species and decided to proceed only with $ACD_{Mg/Ca}$ for all species. For *G. truncatulinoides* there were also large deviations in ACD estimates between methods especially in the northern hemisphere. Perhaps these may indeed be due to degree of gametogenic calcite in the specimens of the different regions. Since the reviewer brought this up, and *G. truncatulinoides* is potentially vital effects free and we already had the data we included these extra statistical analyses in the supplements.

The narrative does not change much. Note that $ACD_{Mg/Ca}$ and ACD_{iso} may affect the relationships with *in situ* oceanographic parameters (as they are derived from considerably different depths) it does not affect the correlation between geochemically derived densities that we calculated based on the combined Mg/Ca and isotopic data.

Minor points:

- Line 18: could the authors be more specific than “part of the surface carbon”?

Thank you for clarification. We actually meant “sea” surface carbon.

- Line 30: bodies aren't enclosed in the shells – cytoplasm also flows outside of the shell

This is indeed a subtle point of wording. We originally used the term “enclose” with the freedom of interpretation that foraminifera *have the potential to* enclose their body within the shell. However, to enhance biological accuracy and interpretive clarity, we have revised the sentence to use 'house' instead. This phrasing offers perhaps greater flexibility and may better reflect the fact that living foraminifera can retract into their shell when needed, while also allowing freedom to imply that the cytoplasm can emerge beyond the shell.

- Line 32: Sorry, this is a bit pedantic, but there are many active swimmers that are still generally considered planktonic rather than nektonic (e.g., copepods) in part due to their size and relatively small distances travelled despite active swimming behaviors

We have removed the parenthetical clarification.

- Line 193: “This is similar to” is free floating.

Corrected

- Please make the table 2 caption more descriptive? It is difficult to interpret several of these columns

Explanatory text has now been added to the caption for clarity.

- I would suggest adding a figure of surface density through the study area (and perhaps temperature and salinity?)

Figure 1 was redrawn to add the additional parameters requested.

Citations:

Reynolds, C. E., Richey, J. N., Fehrenbacher, J. S., Rosenheim, B. E., & Spero, H. J. (2018). Environmental controls on the geochemistry of *Globorotalia truncatulinoides* in the Gulf of Mexico: Implications for paleoceanographic reconstructions. *Marine Micropaleontology*, 142, 92-104.
Davis, C. V., Fehrenbacher, J. S., Hill, T. M., Russell, A. D., & Spero, H. J. (2017). Relationships between temperature, pH, and crusting on Mg/Ca ratios in laboratory-grown *Neogloboquadrina*

foraminifera. *Paleoceanography*, 32(11), 1137-1152.

Hupp, B. N., & Fehrenbacher, J. S. (2023). Geochemical differences between alive, uncrusted and dead, crusted shells of *Neogloboquadrina pachyderma*: Implications for paleoreconstruction.

Paleoceanography and Paleoclimatology, 38(10), e2023PA004638.

Reviewer #1 (Remarks to the Author):

The manuscript has been well revised. However, could you please revise Figure 4 more clearly visible? It is difficult to see the letters, numbers, and the "C" and "D" in the lower left corner of each graph. If the above minor point is adequately addressed, I think this manuscript will be acceptable for publication in Communications Earth & Environment.

Thank you. Figure 4 has been revised accordingly.

Reviewer #2 (Remarks to the Author):

I am satisfied that all of my comments have been fully addressed. I will add just two minor suggestions around language.

Thank you

Line 65: deeper -> subthermocline depths (or other more descriptive term)

Changed

Line 72: "total virtual "cell" volume" -> internal shell volume

Changed to "cell" total volume as we are not only considering the internal volume but the volume if the shell itself.